# Impervious Surface Mapping Based on Remote Sensing and an Optimized Coupled Model: The Dianchi Basin as an Example

**Yimin Li [1,2,†], Xue Yang [1,*,†], Bowen Wu [1], Juanzhen Zhao [3] and Xuanlun Deng [1]**

[1] School of Earth Sciences, Yunnan University, Kunming 650091, China; liym@ynu.edu.cn (Y.L.); a460929153@163.com (B.W.); dengxl@mail.ynu.edu.cn (X.D.)

[2] Yunnan Provincial University Domestic High Score Satellite Remote Sensing Geological Engineering Research Center, Kunming 650091, China

[3] Institute of International Rivers and Ecological Security, Yunnan University, Kunming 650091, China; zjz1998@mail.ynu.edu.cn

\* Correspondence: yx9121@mail.ynu.edu.cn

† These authors contributed equally to this work.

**Abstract:** Accurately extracting impervious surfaces (IS) and continuously monitoring their dynamics are crucial practices for promoting sustainable development in regional ecological environments and resources. In this context, we conducted experiments to extract IS of the Dianchi Lake Basin by utilizing various features extracted from remote sensing images and applying three different machine learning algorithms. Through this process, we obtained the optimal combination of features and a machine learning algorithm. Utilizing this model, our objective is to map the evolution of IS in the Dianchi Lake Basin, from 2000 to 2022, and analyze its dynamic changes. Our results showed the following: (1) The optimal model for IS extraction in the Dianchi Lake Basin was IMG-SPESVM based on the support vector machine, remote sensing images, and spectral features. (2) From 2000 to 2022, the spatial distribution and shape of the IS in the Dianchi Lake Basin changed significantly, but they all developed in the area around Dianchi Lake. (3) From 2000 to 2015, the rate of expansion of IS gradually accelerated, while from 2015 to 2022, it contracted. (4) From 2000 to 2022, the center of mass of IS moved to the northeast, and the standard deviation ellipse shifted greatly in the south–north direction. (5) Natural factors negatively affected the expansion of IS, while social factors positively affected the distribution of the IS.

**Keywords:** impervious surface; Dianchi Basin; machine learning algorithm; model optimization

## 1. Introduction

The expansion of impervious surfaces can drive global land cover and land use change, and it is a result of global economic growth and environmental changes [1]. Impervious surface (IS) is a type of surface coverage where water cannot infiltrate below the surface layer, and it mainly includes artificial landscapes, such as roads, squares, parking lots, and building tops [2]. As an artificial land cover, the IS strongly affects the quality of the regional ecological environment. The proliferation of IS mirrors urbanization patterns, and it can be viewed as a tangible result of economic globalization. Economic globalization instigates a redistribution of the urban populace, prompting extensive and rapid urban transformation. These changes, in turn, alter the composition and properties of the land cover beneath, correlating with an elevated demand for ecological and environmental resources. Over recent decades, global economic growth has yet to disengage from ecological demand [3]. For developing nations, such as China, certain disparities in urbanization exacerbate the imbalance between economic and ecological costs [4], leading to numerous environmental and ecological issues. Examples include an increased risk of urban flooding [5], the intensification of the urban heat island effect [6], and the diminishment of carbon storage and biodiversity [7,8]. Hence, comprehending the current state and expansion mechanisms

of IS, along with studying the rules governing the dynamic change of IS, is crucial for sustainable urban ecological development.

Traditional methods for identifying IS largely rely on data collection, field surveys, or mapping. However, these approaches are labor-intensive, time-consuming, slow in updating data, and exhibit poor real-time performance. Rapidly advancing remote sensing technology is now widely utilized for the mapping and dynamic monitoring of IS, given its convenience, speed, and real-time capabilities [9]. Existing IS extraction methods, based on remote sensing technology, primarily comprise the regression model method [10], the exponential method [11], the spectral mixture–decomposition method [12], and various machine learning methods [13]. The exponential method is preferred by many researchers for its simplicity and ease of implementation, but its subjective threshold selection and the spatial resolution limitations of remote sensing images may lead to the misclassification of ground objects [14]. The spectral mixture–decomposition method is more objective; however, the selection of IS endmembers is intricate, and an improper selection can significantly affect the precision of IS extraction [15].

Machine learning algorithms excel in detecting changes in land use and land cover. Ghayour et al. [16] evaluated the performance of several machine learning algorithms for generating land use and land cover (LULC) maps using Sentinel 2 and Landsat 8 satellite data, wherein the overall accuracy of the support vector machine classifier was reported to be 94%. Based on the GEE platform, Saeid et al. [17] mapped LULC variations accurately by using historical Landsat datasets, demonstrating the efficacy of the random forest algorithm as a potent classifier. Machine learning algorithms are also progressive techniques in the investigation of artificial surface extraction. These models minimize the influence of subjective factors in the learning process, providing swift and precise recognition. For instance, Mahyou et al. [18] used the random forest algorithm to extract sample points and artificial neural networks to identify IS, which effectively and accurately extracted the impervious water surface of Marrakesh. Utilizing Sentinel 1 and Sentinel 2 data, Shrestha et al. [19] employed the random forest algorithm to identify the IS of nine cities in Pakistan, achieving an overall classification accuracy between 85% and 98%. Esch et al. [20] combined Landsat images and road network data, with the support vector machine method, to map IS in parts of Germany, attesting to the method's ability to accurately map large-scale IS. Jiang et al. [21] improved the extraction accuracy of Baoding's built area by using Landsat 8 images and night lighting data, along with the support vector machine algorithm. Despite their significant role in land use and artificial surface monitoring, machine learning algorithms encounter limitations when applied to IS recognition in medium-resolution remote sensing images. The primary reason lies in the inherent complexity and computational intensity of machine learning algorithms. Additionally, medium-resolution remote sensing images are inherently constrained by limitations in resolution and imaging performance. When these limitations are combined with the rich diversity and intricate complexity of the landscape, the result is spectral confusion. This, in turn, leads to a reduction in the accuracy of impervious water extraction [22].

Many researchers have fused multiple features of remote sensing images to obtain high-quality information on IS and improve the accuracy of IS extraction. Shaban et al. [23] extracted three texture features, including the gray level co-occurrence matrix (GLCM), gray level difference histogram (GLDH), and difference histogram (SADH), which were combined with spectral features to significantly improve the accuracy of IS extraction. Wang et al. [24] used the normalized difference vegetation index (NDVI) time series, reflectance spectral features, and spatial texture features as the feature input of support vector machine classification and extracted IS. The overall classification accuracy of the method was 93.66%. The IS extraction method, based on multi-feature inputs, can significantly improve spectral mixing and increase the classification accuracy of IS. Traditional research methodologies typically take into account a limited number of features present in remote sensing images, thereby failing to comprehensively capture the distinctions between IS and other land types. There are studies that attempt to leverage a diverse set of features in the process

of IS identification, but they often overlook the significance of feature selection [25,26], leading to feature redundancy. In this research, it is hypothesized that the classification accuracy may be influenced by both the category and the number of combined features, and distinct categories of features might be optimally suited to varying machine learning algorithms. Consequently, we selected spectral features, texture features, and seasonal land cover features for further examination. This was done to investigate the extraction accuracy of IS under the impact of varying feature combinations and diverse machine learning algorithms.

The Dianchi Lake Basin is the largest plateau lake basin in the Yunnan–Guizhou Plateau. It is an important water conservation ecological function area, identified based on the ecosystem assessment and ecological security of China [27], and it is also the most economically dynamic area in Yunnan Province. Due to the rapid expansion of IS, the urban surface of the Dianchi Lake Basin has undergone rapid and dramatic changes, threatening the regional ecological environment, which is vulnerable and sensitive. Therefore, in this study, we investigated the Dianchi Lake Basin, based on remote sensing and machine learning algorithms, to realize the following objectives: (1) To optimize the IS extraction method, the optimal coupling model of the machine learning algorithm and remote sensing features are selected, and they map the IS in the Dianchi Lake Basin from 2000 to 2022. (2) Based on the long time-series mapping results of IS, the dynamic change characteristics of IS in the Dianchi Lake Basin were quantitatively analyzed. Furthermore, building upon the foundation of prior studies, we implemented two key enhancements: (1) We carried out comparisons among the coupling models of various machine learning algorithms and different remote sensing features. The aim was to select the most effective coupling model to maximize the accuracy of IS extraction. (2) We adopted the empirical analysis methodology of the partial least square structural equation model. This was done to dissect the causative link between IS distribution and its influencing factors, thereby providing a more precise depiction of the impact of both latent and observed variables.

## 2. Study Area and Data

### 2.1. Study Area

The Dianchi Lake Basin (Figure 1) is located in the middle of the Yunnan Plateau (102°29′–103°10′ E, 24°29′–25°28′ N). The area has a relatively independent ecosystem that includes the Dianchi Lake at the center and the water systems flowing into the Dianchi Lake. The Dianchi Lake Basin is an important area of urbanization in Kunming. It mainly consists of seven districts and counties [28], including the Wuhua District, the Xishan District, the Panlong District, the Guandu District, the Chenggong District, the Jinning District, and the Songming County. The total area is about 2900 km$^2$; it includes the main economic sites, and it houses most of the population of Yunnan Province. The basin is higher in the northeast than in the southwest, with an altitude of 1735–2825 m. The region possesses a tropical plateau monsoon climate, characterized by distinct dry and wet periods throughout the year, and it offers a temperature range conducive to human habitation. The wet season spans from May to October, while the dry season extends from November through April. The Dianchi Lake Basin epitomizes a quintessential plateau basin. It has undergone rapid urban expansion and rapid growth of construction lands, such as residential land, tourism land, and transportation facilities. These changes have strongly affected the regional ecosystem structure and land use pattern.

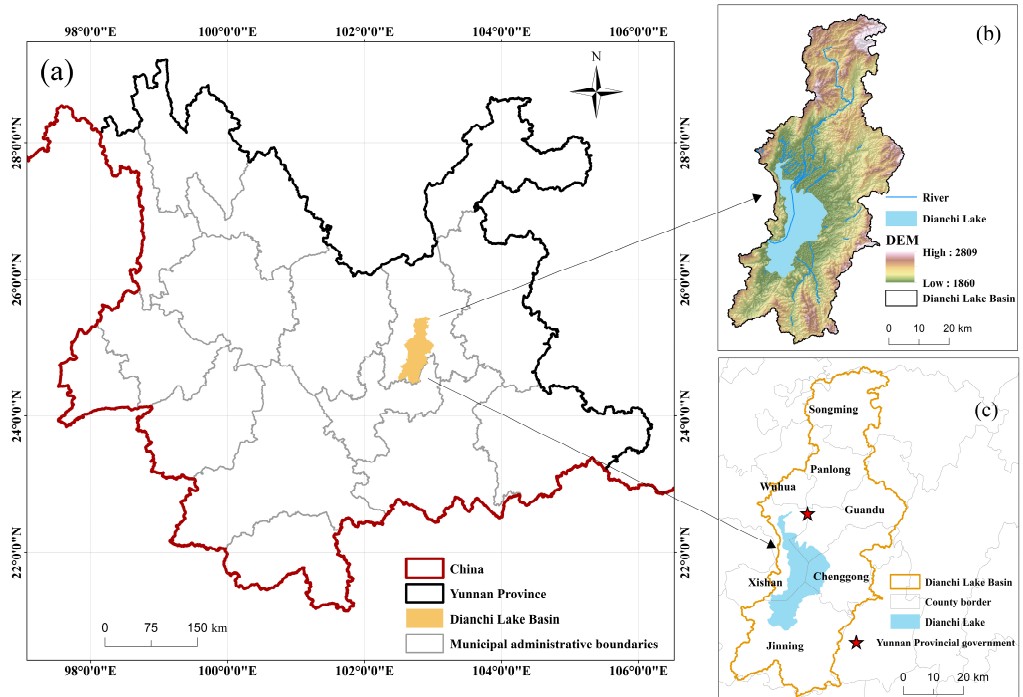

**Figure 1.** Study area. (**a**) shows the location of Dianchi Lake Basin in Yunnan Province. (**b**) shows the elevation of Dianchi Lake basin. (**c**) shows the county-level administrative regions involved in the Dianchi Lake Basin.

*2.2. Data*

The data used in this study mainly included remote sensing data and statistical data (Table 1). (1) Remote sensing data: Landsat remote sensing images, from 2000 to 2022, were used as the data source, and Landsat images with no cloud or low cloud coverage in the study area were selected (see Table 2 for specific data products); all data were obtained from the official website of the United States Geological Survey. The MOD13A1 data used to obtain NDVI were obtained from the official remote sensing data website of NASA. The data on elevation were obtained from the SRTM (Shuttle Radar TopogRaphy Mission) digital elevation model provided by the geospatial data cloud official website. High-resolution historical satellite imagery and land use data were procured from the Bigemap website and the Data Center for Resources and Environmental Sciences under the Chinese Academy of Sciences, respectively. GAIA (Global artificial impervious area) IS products were extracted from the Finer Resolution Observation and Monitoring-Global Land Cover-2015 v0.1. (2) Additionally, statistical data were gathered, which include annual average temperature, annual average rainfall, and GDP data. These were sourced from the Data Center for Resources and Environmental Sciences of the Chinese Academy of Sciences and the Yunnan Statistical Yearbook. The data on A-level scenic spots were obtained from the Yunnan Provincial Department of Culture and Tourism. The data on the population and road network were obtained from the official websites of WoldPop and OpenStreetMap.

**Table 1.** Data and sources.

| Type | Name | Time Range | Data Source | Usage |
|---|---|---|---|---|
| Remote Sensing data | Landsat | 2000–2022 | United States Geological Survey official website (https://www.usgs.gov/) | Extract spectral features and texture features, select samples, extract IS, calculate remote sensing ecological index |
| | MOD13A1 | 2022 | NASA official remote sensing data network (https://ladsweb.modaps.eosdis.nasa.gov/) | Obtain the seasonal features of land cover |
| | Digital elevation model (DEM) | - | Geospatial Data Cloud Official Website (www.gscloud.cn) | As a factor affecting the distribution of IS |
| | High definition satellite images | 2000–2022 | Bigemap (http://www.bigemap.com/) | Used to assist in selecting samples |
| | Land use data | 2000–2022 | Data Center for Resources and Environmental Sciences of the Chinese Academy of Sciences (https://www.resdc.cn/) | Calculate carbon storage |
| Statistical data | Global artificial impervious area (GAIA) | 2000–2018 | http://data.ess.tsinghua.edu.cn/ | Evaluate the extraction accuracy of IS |
| | A-level scenic spot catalogue | 2010–2022 | Yunnan Provincial Department of Culture and Tourism official website (http://dct.yn.gov.cn/) | |
| | Mean annual temperature | 2000–2022 | Data Center for Resources and Environmental Sciences of the Chinese Academy of Sciences (https://www.resdc.cn/)Yunnan Statistical Yearbook | As factors affecting the distribution of IS |
| | Average annual rainfall | 2000–2022 | | |
| | GDP | 2000–2022 | | |
| | Population | 2000–2022 | Woldpop (https://www.worldpop.org/) | |
| | Roads data | 2000–2010 | Openstreetmap (https://www.openstreetmap.org/) | |

**Table 2.** Landsat data products.

| Data Set | Spatial Resolution | Time Range |
|---|---|---|
| Landsat 5 TM | 30 m | 20000212 20050209 20101224 |
| Landsat 8 OLI_TIRS | | 20150104 20220312 |

## 3. Methodology

The study had two main parts (Figure 2). (1) IS extraction: Based on different combinations of spectral features, texture features, and seasonal features of surface coverage, the optimal coupling model of IS extraction was selected after coupling with three machine learning algorithms (artificial neural network, support vector machine, and random forest). The IS of the Dianchi Lake Basin, from 2000 to 2022, was extracted based on the optimal coupling model. (2) Dynamic change analysis of IS: Based on the spatial and temporal distribution of IS, from 2000 to 2022, the expansion speed, expansion direction, spatial correlation, and driving mechanism of IS were studied, and the dynamic changes in IS in the Dianchi Lake Basin were evaluated.

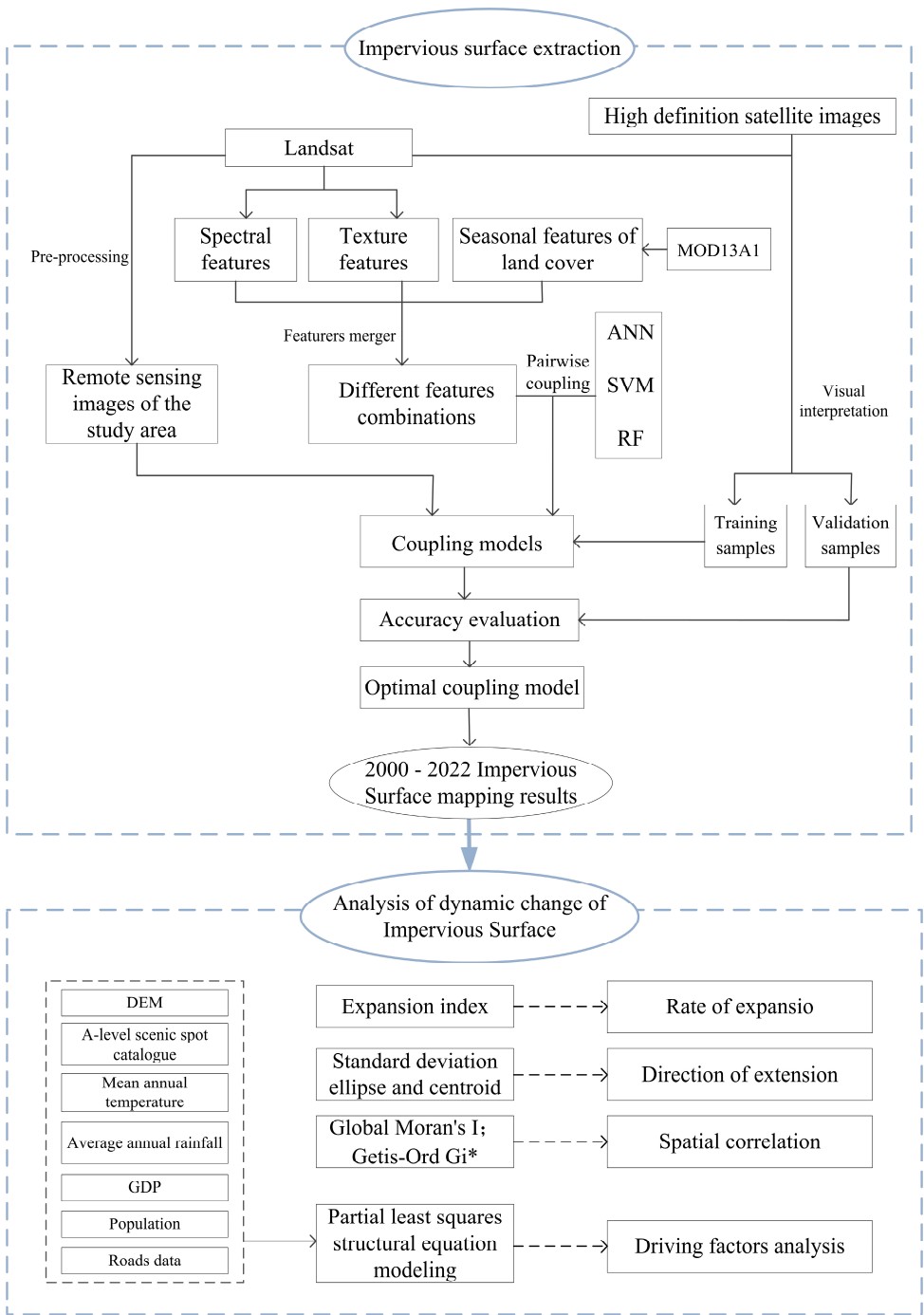

**Figure 2.** Research framework.

*3.1. Impervious Surface (IS) Extraction*

3.1.1. Machine Learning Algorithm

Classification based on machine learning algorithms is commonly performed for IS extraction. Specifically, artificial neural networks, support vector machines, and random forests are widely used in IS extraction studies, as these algorithms have high recognition accuracy. Therefore, we selected the three machine learning algorithms, discussed above, for IS extraction experiments in the Dianchi Basin.

(1)　Artificial neural network (ANN)

An artificial neural network (ANN) is a computational system that is designed in accordance with the biological neural network structure of the animal brain. It boasts a

suite of advantages, including nonlinearity, a robust resistance to disturbance, and high adaptability [29]. The training of an ANN involves the processing of instances, with each comprising a known "input" and "result". The learning rules are then adjusted based on the discrepancy between the "processing output" and the "target output". ANNs possess formidable modeling capacities, particularly when the relationship within the underlying data remains unidentified.

An ANN constitutes an assembly of interlinked units and nodes that represent artificial neurons, devised to emulate neurons in a biological brain. It operates by receiving impulses from other neurons, transmitting these impulses as outputs, and subsequently forwarding them to other neurons in a cyclical manner. Neurons are real functions of the input vector $(x_1, \ldots, x_n)$, and the output function is shown below [30]:

$$f(y) = f\left(\theta + \sum_{i=1}^{n} w_i \times x_i\right) \tag{1}$$

Here, $f$ is an activation function, and $\theta$ represents the threshold value.

(2) Support vector machine (SVM)

A support vector machine (SVM) represents a supervised learning algorithm and serves as a classification prediction model that operates on statistical principles [31]. Generally speaking, its fundamental principle involves assigning a group of training samples into two categories. After the modeling of the SVM's training algorithm, the two types of samples are allocated to their respective models for training, eventually becoming a non-probabilistic binary linear classifier used for classification, regression analysis, or anomaly detection. Besides addressing linear classification problems, SVMs are also proficient at performing nonlinear classification. Often hailed as one of the most resilient prediction methods, SVMs are capable of tackling a multitude of practical issues.

Assuming that the n-dimensional space training sample set is $\{X_i, Y_i\}$ $[i = 1, 2, \ldots, n]$ ($n$ represents the number of samples), a linear regression function can be expressed as follows [32]:

$$f(x) = \omega \varphi(x) + b \tag{2}$$

Here, $\omega$ represents the direction vector, $\varphi(x)$ represents the mapping function, and $b$ represents the bias term. The problem of solving $\omega$ and $b$ can be transformed into the problem of finding the extremum of the objective function as follows:

$$min\left(\frac{1}{2}||\omega||^2 + c\sum_{i=1}^{n}(\xi i + \xi i^*)\right) \tag{3}$$

$$s.t\begin{cases} f(x_i) - y_i \leq \varepsilon + \xi i \\ y_i = f(x_i) \leq \varepsilon + \xi i^* \\ \xi i \geq 0; \xi i^* \geq 0; i = 1, 2, \ldots, n \end{cases} \tag{4}$$

Here, $c$ represents the penalty factor; $\xi i$ and $\xi i^*$ represent the relaxation variables; $\varepsilon$ represents the loss function. The Lagrangian multipliers $a_i$ and $a_i^*$ are introduced, and the Lagrangian function is constructed to obtain the pairwise form:

$$maxQ(a, a^*) = \frac{1}{2}\sum_{i=1}^{k}\sum_{j=1}^{k}(a_i - a_i^*)\left(a_j - a_j^*\right)(x_i - x_j) - \varepsilon\sum_{i=1}^{k}(a_i + a_i^*) + \sum_{i=1}^{k}(a_i + a_i^*)y_i \tag{5}$$

$$\sum_{i=1}^{k}(a_i - a_i^*) = 0(a_i \geq 0, a_i^* \leq C, i = 1, 2, \ldots, n) \tag{6}$$

Solving the above equation yields the SVM regression function:

$$f(x) = \sum_{i=1}^{n}(a_i - a_i^*)K(x_i, x_j) + b \tag{7}$$

Here, $K(x_i, x_j) = \varphi(x_i)\varphi(x_j)$ is the inner product kernel function, which the SVM maps use to sample to a higher dimensional space *H*. It also performs a linear partitioning function of the original problem in *H*.

(3)     Random Forest (RF)

Random Forest (RF) constitutes an ensemble algorithm, which is well-suited for tasks involving classification, regression, and prediction. It incorporates numerous decision trees, with the output category being determined by the count of categories produced by an individual tree [33]. Its advantage is that the random sampling process and the features of the random forest decrease the sensitivity to data noise and outliers in the classification process, thus avoiding overfitting [34]. A sequence of classification models $\{h_1(X), h_2(X) \ldots h_k(X)\}$ is obtained after training through *K* rounds. This sequence constitutes a multi-classification model system, and the final classification results are obtained by using a simple majority voting decision [35]. The final classification decision is shown in Equation (8).

$$H(x) = argmax_Y \sum\nolimits_{i=1}^{k} I(h_i(x) = Y) \tag{8}$$

Here, $H(x)$ denotes the combined classification model, $h_i$ denotes the individual decision tree classification model, *Y* denotes the output variable, and *I* denotes the trialability function.

### 3.1.2. Remote Sensing Image Multi-Feature Extraction

The reduction in spectral confusion is the key to extracting IS at the pixel scale. However, distinguishing IS from other ground objects using only a single feature is difficult, and it limits the accuracy of IS extraction. In this study, the spectral features, texture features, and seasonal features of the ground cover of remote sensing images were extracted from multi-temporal remote sensing images (Table 3). The specific features are presented as follows:

(1)     Spectral features (SPE): Algebraic operations between different spectral bands can highlight the spectral features of remote sensing images. In this study, the normalized difference IS index (ENDISI) and the modified normalized difference water body index (MNDWI) were used to distinguish between IS and water bodies [36]. Additionally, the minimum noise fraction (MNF) image enhancement method was used to improve the signal-to-noise ratio of the image and enhance the discrimination between different ground cover types [37]. The first three components of MNF were used as an aid to extract the spectral features of IS.

(2)     Texture features (TEX): Most IS are anthropogenic, exhibiting regular geometric patterns that differentiate their texture features from those of other entities within an image. These texture attributes can be employed to accentuate the information pertaining to IS [38]. In this investigation, the Gray Level Co-occurrence Matrix (GLCM) was utilized to extract such texture features [39].

(3)     Seasonal features of ground cover (SSC): A major difficulty in IS extraction is that IS are easily confused with bare soil. The IS is almost impervious to the seasonal changes, while bare soil areas tend to have differences in vegetation cover due to different seasons, so this feature can be used to accurately differentiate between IS and bare soil. Therefore, the normalized vegetation index (NDVI) of four seasons, including spring, summer, autumn, and winter, was extracted as the seasonal features of surface cover for extracting IS [36].

**Table 3.** Remote sensing image features.

| Feature Type | Feature Number | Band Name | Meaning of Band |
|---|---|---|---|
| Spectral signature | 5 | ENDISI, MNDWI, MNF1, MNF2, MNF3 | MNF1, MNF2, and MNF3 are the first three components of MNF |
| Texture feature | 8 | Mean, Variance, Homogeneity, Contrast, Dissimilarity, Entropy, Second Moment, Correlation | Geometric features |
| Seasonal features of land cover | 4 | NDVI1, NDVI2, NDVI3, NDVI4 | NDVI1–4 is NDVI in spring, summer, autumn and winter |

As different features might contain the same information, and the class and number of fused features might affect the extraction accuracy, different features were combined as follows: Combination 1 was based on spectral features; Combination 2 was based on texture features; Combination 3 was based on ground cover seasonal features; Combination 4 was based on spectral features and texture features; Combination 5 was based on spectral features and ground cover seasonal features; Combination 6 was based on texture features and ground cover seasonal features; Combination 7 was based on spectral features, texture features, and ground cover seasonal features.

3.1.3. Accuracy Evaluation Methods

(1)    Accuracy evaluation based on pixels

The sample points of IS and other land use types in the study area were collected using high-definition historical satellite images, land use data, and Landsat images. Then, they were evaluated based on the overall accuracy (OA), the Kappa coefficient, producer accuracy (PA), and user accuracy (UA). The calculation formulas are as follows:

$$OA = \frac{1}{N} \sum_{i=1}^{r} x_{ii} \tag{9}$$

$$Kappa = \frac{N \sum_{i=1}^{r} x_{ii} - \sum_{i=1}^{r} (x_{i+} \times x_{+i})}{N^2 - \sum_{i=1}^{r} (x_{i+} \times x_{+i})} \tag{10}$$

where $N$ is the total number of pixels, $x_{ii}$ is the diagonal element of the confusion matrix, and $x_{i+}$ and $x_{+i}$ are the sum of rows and columns, respectively.

$$UA = \frac{TP}{TP + FP} \tag{11}$$

$$PA = \frac{TP}{TP + FN} \tag{12}$$

where *TP* or *TN* is the number of "IS" or "non-IS" that is correctly classified, *FP* is the number of "non-IS" pixels misclassified as "IS", and *FN* is the number of "IS" pixels misclassified as "non-IS".

(2)    Accuracy evaluation based on indices

The consistency and fit between the results of the classification of IS and available products (GAIA data) can be assessed to evaluate the accuracy. The percentage of IS in each

image element was calculated by regional statistics, and the root mean square error (RMSE) and coefficient of determination ($R^2$) were used as judgment indices for evaluation.

$$RMSE = \sqrt{\frac{\sum_{i=1}^{n}(y_i - \hat{y}_i)^2}{n}} \tag{13}$$

$$R^2 = 1 - \frac{\sum_{i}^{n}(\hat{y}_i - y_i)^2}{\sum_{i}^{n}(\overline{y_i} - y_i)^2} \tag{14}$$

In the formula, $n$ is the number of samples, and $y_i$, $\hat{y}_i$, and $\overline{y_i}$ are the real value, the predicted value, and the real value, respectively.

### 3.1.4. Selection of the Optimal Coupling Model

To select the optimal coupled model for IS extraction in the Dianchi Basin, 24 sets of experiments were set up for the study based on the data collected in 2022 (Table 4). In the experiments, to consider the land use cover categories and remote sensing image characteristics, the features were divided into IS, vegetation, water bodies, and bare land (the latter three were all permeable). The sample points were randomly and evenly selected within the study area (Figure 3), a total of 8050 pixels were selected as the IS sample points, and 19,145 pixels were selected as the permeable surface sample points. About 70% of the sample points were used as training samples for model training, and 30% were used as validation samples.

**Table 4.** Introduction of Impervious Surface extraction experiments.

| Machine Learning Algorithm | Input Layers | The Abbreviation of the Experiments |
|---|---|---|
| ANN | Remote Sensing image | IMG$^{ANN}$ |
| | Remote Sensing image, spectral features | IMG-SPE$^{ANN}$ |
| | Remote Sensing image, seasonal features of land cover | IMG-SSC$^{ANN}$ |
| | Remote Sensing image, seasonal features of land cover, spectral features | IMG-SSC-SPE$^{ANN}$ |
| | Remote Sensing image, texture features | IMG-TEX$^{ANN}$ |
| | Remote Sensing image, spectral features, texture features | IMG-SPE-TEX$^{ANN}$ |
| | Remote Sensing image, seasonal features of land cover, texture features | IMG-SSC-TEX$^{ANN}$ |
| | Remote Sensing image, spectral features, seasonal features of land cover, texture features | IMG-SPE-SSC-TEX$^{ANN}$ |
| SVM | Remote Sensing image | IMG$^{SVM}$ |
| | Remote Sensing image, spectral features | IMG-SPE$^{SVM}$ |
| | Remote Sensing image, seasonal features of land cover | IMG-SSC$^{SVM}$ |
| | Remote Sensing image, seasonal features of land cover, spectral features | IMG-SSC-SPE$^{SVM}$ |
| | Remote Sensing image, texture features | IMG-TEX$^{SVM}$ |
| | Remote Sensing image, spectral features, texture features | IMG-SPE-TEX$^{SVM}$ |
| | Remote Sensing image, seasonal features of land cover, texture features | IMG-SSC-TEX$^{SVM}$ |
| | Remote Sensing image, spectral features, seasonal features of land cover, texture features | IMG-SPE-SSC-TEX$^{SVM}$ |
| RF | Remote Sensing image | IMG$^{RF}$ |
| | Remote Sensing image, spectral features | IMG-SPE$^{RF}$ |
| | Remote Sensing image, seasonal features of land cover | IMG-SSC$^{RF}$ |
| | Remote Sensing image, seasonal features of land cover, spectral features | IMG-SSC-SPE$^{RF}$ |
| | Remote Sensing image, texture features | IMG-TEX$^{RF}$ |
| | Remote Sensing image, spectral features, texture features | IMG-SPE-TEX$^{RF}$ |
| | Remote Sensing image, seasonal features of land cover, texture features | IMG-SSC-TEX$^{RF}$ |
| | Remote Sensing image, spectral features, seasonal features of land cover, texture features | IMG-SPE-SSC-TEX$^{RF}$ |

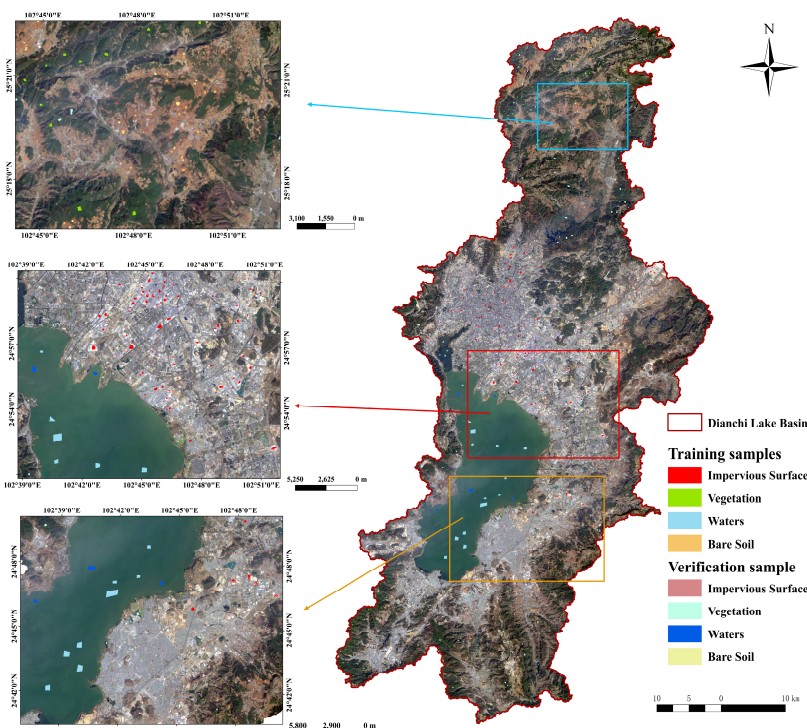

**Figure 3.** Distribution of sample points of the IS extraction experiment in 2022.

The ANN, SVM, and RF algorithms were executed on the ENVI platform, with their parameters set in accordance with the ENVI user manual. For the ANN algorithm, the Activation function chosen was the Logistic function. The Training Threshold Contribution was employed to modulate the shift in the internal weight of the node, with the parameter being designated as 0.9. The Training Rate was established at 0.2, while the Training Momentum, which facilitated weight alteration along the current direction, was designated as 0.9. The training concluded when the RMS error reached 0.1. The Number of Hidden Layers was set at 1, and the Number of Training Iterations was established at 1000. Regarding the SVM algorithm, the Kernel Type selected was the Radial Basis Function, the Gamma Index was set at 0.143, and the default Penalty Parameter was 100. The RF algorithm invoked the extension tool of the ENVI platform, with the Number of Trees fixed at 100 and the Gini Index being used as the Impurity Function. The Minimum Number of Samples was set to 1.

The confusion matrix and the results of the evaluation of the 24 IS datasets' accuracy were determined (Figure 4), and the optimal coupling model for IS extraction in the Dianchi Basin was obtained as IMG-SPE$^{SVM}$.

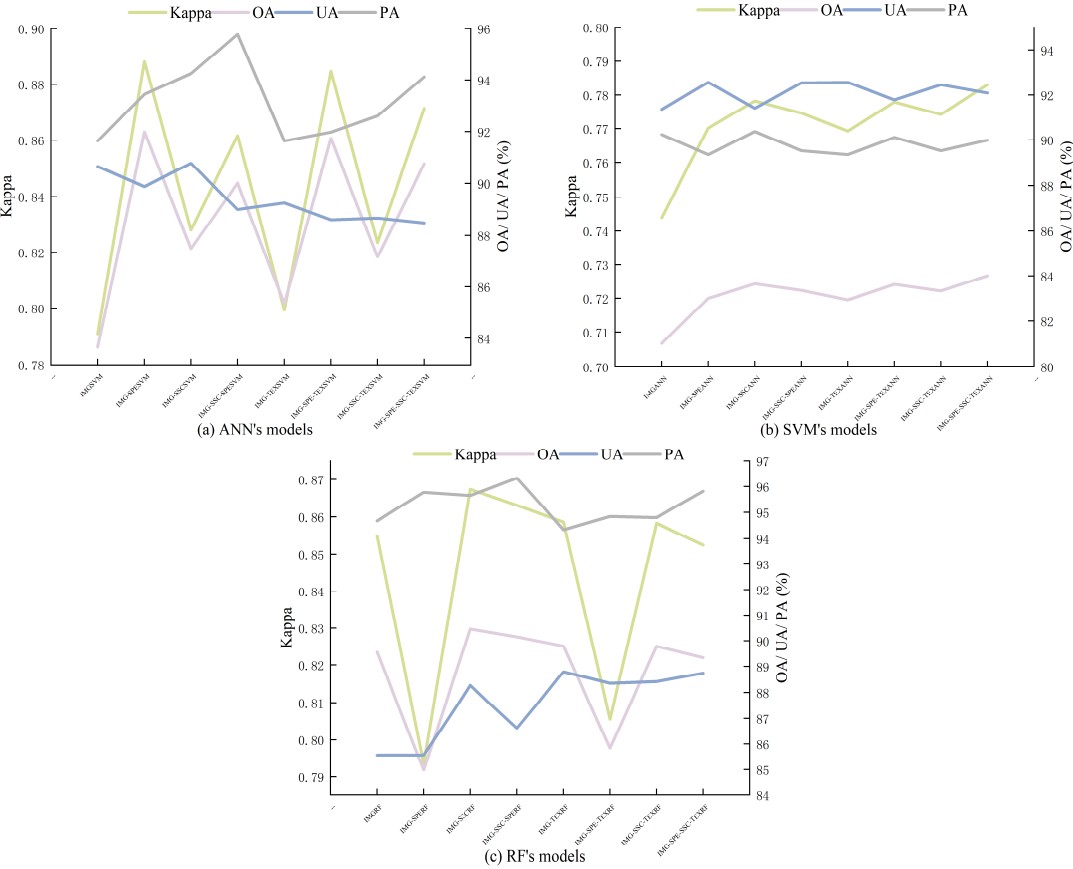

**Figure 4.** Classification accuracy of each coupling model.

### 3.2. Dynamic Change Analysis of IS

3.2.1. The IS Expansion Index

The rate of IS expansion is the growth of IS within a specific duration. It can be used to quantitatively assess the rate of change of IS within a specific duration, expressed as the expansion index [40]:

$$V = \frac{U_b - U_a}{T} \tag{15}$$

Here, $V$ indicates the IS expansion rate index, $T$ indicates the time interval (year), $U_a$ indicates the IS area at the beginning of the study, and $U_b$ indicates the IS area at the end of the study. The expansion rate can be classified as low expansion ($V \leq 10$), medium expansion ($10 < V \leq 20$), fast expansion ($20 < V \leq 50$), and high expansion ($V > 50$).

3.2.2. Analysis of the Standard Deviation Ellipse and the Center of Mass

The standard deviation ellipse and the center of mass are spatial statistical models that can be used to accurately determine the centrality, directionality, and deviation of change direction of the spatial distribution of geospatial elements. We used it to analyze the directional characteristics of IS expansion in the Dianchi Basin; the azimuthal short axis indicated the degree of dispersion, and the long axis reflected the directionality of the spatial distribution [41], which was calculated as follows:

$$M = \left( \sqrt{\frac{\sum_1^n x_i}{n}}, \sqrt{\frac{\sum_1^n y_i}{n}} \right) \tag{16}$$

$$D = \sqrt{\frac{\sum_1^n (x_i \cos R - y_i \sin R)^2}{\sum_1^n (x_i \sin R - y_i \cos R)^2}} \tag{17}$$

$$S = \sqrt{\frac{\sum_1^n (x_i sinR - y_i cosR)^2}{n}} \tag{18}$$

$$R = tan^{-1} \left[ \frac{\sum_1^n x_i{}^2 - \sum_1^n y_i{}^2 + \sqrt{\left(\sum_1^n x_i{}^2 - \sum_1^n y_i{}^2\right)^2 + 4\sum_1^n x_i y_i}}{2\sum_1^n x_i y_i} \right] \tag{19}$$

Here, *M* represents the center point coordinates of the standard deviation ellipse, $x_i$ and $y_i$ represent the two-dimensional spatial coordinates of the *i*th geographic element, n represents the number of geographic elements, *D* represents directionality, *S* represents dispersion, and *R* represents the azimuth of the standard deviation ellipse.

### 3.2.3. Analysis of Spatial Correlation

Spatial autocorrelation refers to regions with similar locations with similar values of variables, and it is generally divided into global spatial autocorrelation and local spatial autocorrelation. Global spatial autocorrelation describes the spatial characteristics of the attribute values (IS coverage) of the whole region, and it is generally measured using global Moran's I. However, global spatial autocorrelation ignores the local instability in a small area, so local spatial autocorrelation needs to be used to accurately determine the heterogeneous characteristics of spatial elements [42]. In this study, Getis-Ord Gi* was used to analyze the local spatial autocorrelation of IS cover in the Dianchi Basin. The formula used for calculating it is shown below.

$$G_i^*(d) = \sum_{i=1}^n w_{ij}(d) X_j / \sum_{j=1}^n X_j \tag{20}$$

Here, *d* represents the distance scale, and $w_{ij}(d)$ represents the spatial weight between statistical units *i* and *j*.

### 3.2.4. Analysis of PLS-SEM-Based Driving Mechanism

A structural equation model (SEM) is a statistical analysis method that encompasses factor analysis and path analysis. This technique primarily bifurcates into covariance-based structural equation modeling (CB-SEM) and partial least squares-based structural equation modeling (PLS-SEM) [43]. PLS-SEM has fewer sample requirements, and it does not necessitate that the sample data adhere to a normal distribution [44]. In this investigation, the factors influencing the expansion of IS in the Dianchi Basin were scrutinized for employing partial least squares structural equation modeling (PLS-SEM) [45].

With *k* latent variables, there are *k* groups of explicit variables, each containing *m* variables, and each group of explicit variables can be expressed as shown below:

$$X_i = \left\{ X_{i1}, X_{i2}, X_{i3}, \dots, X_{im}{}^i \right\} i = \{1, 2, 3, \dots, k\} \tag{21}$$

The prediction model equation is:

$$X_{ij} = \lambda_{ij} \xi_i + \sigma_{ij} (i = 1, 2, 3, \dots, k; j = 1, 2, 3, \dots, m_i) \tag{22}$$

The equation of the structural model is:

$$\xi_i = \sum_{i \neq j} \beta_{ij} \xi_j + \varepsilon_i \tag{23}$$

Here, $\xi_i$ represents the latent variable after normalization, $\lambda_{ij}$ represents the factor loading, $\beta_{ij}$ represents the path coefficient, and both $\sigma_{ij}$ and $\varepsilon_i$ are error correction terms.

## 4. Results

### 4.1. Impervious Surface (IS) Mapping and Accuracy Evaluation in the Dianchi Basin

Based on the optimal coupled model IMG-SPE$^{SVM}$, the IS of the Dianchi Basin was determined from 2000 to 2015, and the classification accuracy of IS was evaluated for each year based on the pixel and index methods (Table 5). Model training samples and validation samples from 2000 to 2015 were updated every 5 years, based on the image element to calculate the confusion matrix. The OA for 2000–2015 was above 88%, the Kappa coefficient was greater than 0.83, and the UA and PA were greater than 80%. To quantitatively evaluate the accuracy of the extraction results, the spatial consistency between IS and GAIA was verified by evaluating the indices for each year, and the results showed that all $R^2$ values were above 0.65 and all values of RMSE were below 0.2. This indicated that the IMG-SPE$^{SVM}$ model might be applied for extracting IS in the Dianchi Basin, and the classification results could meet the needs of subsequent studies.

**Table 5.** Evaluation of IS extraction accuracy from 2000 to 2015.

| Year | OA | Kappa | UA | PA | $R^2$ | RMSE |
|------|------|-------|------|------|------|------|
| 2000 | 91.9949% | 0.8916 | 0.8000 | 0.9346 | 0.6545 | 0.1016 |
| 2005 | 88.1903% | 0.8386 | 0.8896 | 0.9304 | 0.7541 | 0.1006 |
| 2010 | 89.6847% | 0.8537 | 0.8969 | 0.9006 | 0.6675 | 0.1424 |
| 2015 | 91.0225% | 0.8786 | 0.8103 | 0.9678 | 0.6639 | 0.1721 |

The results of IS mapping in the Dianchi Basin, from 2000 to 2022, are shown in Figure 5. In general, the spatial distribution and shape of the IS of the Dianchi Basin changed significantly from 2000 to 2022. From 2000 to 2005, the IS were mainly distributed along the north shore of Dianchi, including the southeastern part of the Wuhua District, the northeastern part of the Xishan District, the southwestern part of the Panlong District, and the southwestern part of the Guandu District. In 2008, Changshui International Airport was built at the center of the Guandu District, and in 2010, universities were established in the Chenggong District, where a university town was established. This strongly promoted further construction of IS, such as roads, and the IS spread to the middle of the Guandu District, the west of the Chenggong District, and the middle of the Jinning District. After 2010, due to rapid urbanization, the IS of the Dianchi Basin continued to expand.

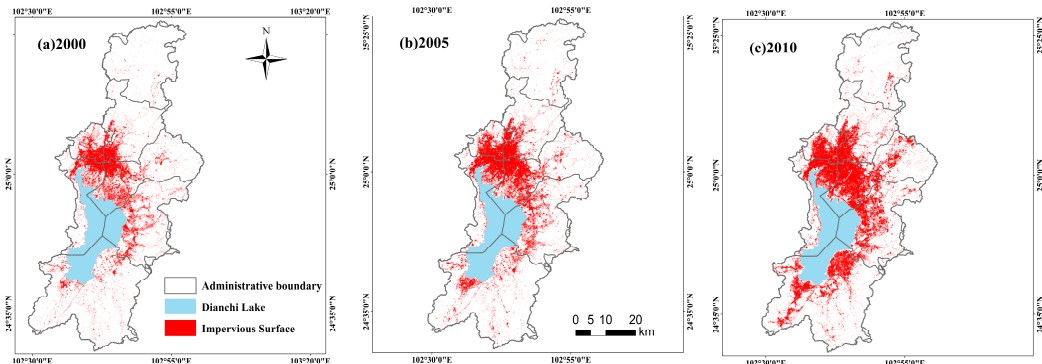

**Figure 5.** *Cont.*

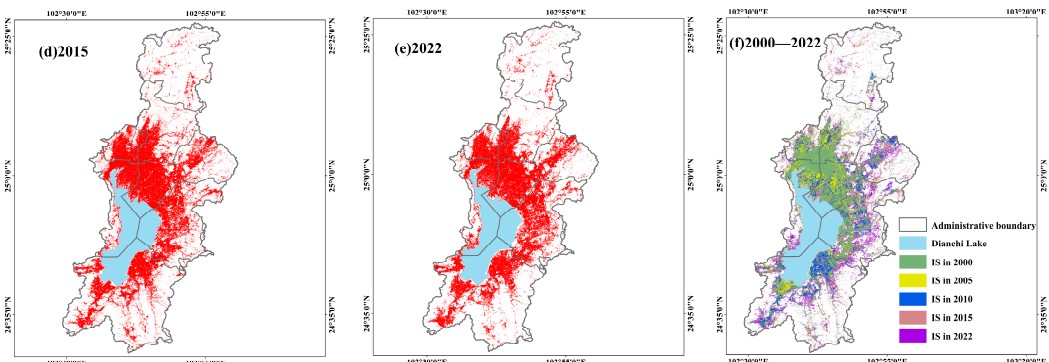

**Figure 5.** The distribution of IS in Dianchi Lake Basin from 2000 to 2022.

*4.2. Analysis of IS Dynamics in the Dianchi Basin*

4.2.1. Rate of Impermeable Surface Expansion

The results of the evaluation of the rate of change of IS in the Dianchi Basin, based on the expansion index, are shown in Table 6. The IS of Dianchi basin expanded at a medium rate from 2000 to 2005 with an expansion area of 86.92 km$^2$, at a fast rate from 2005 to 2010 with an expansion area of 181.02 km$^2$, and at a high rate from 2010 to 2015 with an expansion area of 271.49 km$^2$. The IS expansion from 2000 to 2015 showed sequential acceleration, while that from 2015 to 2022 showed a low rate of contraction. Rapid economic development occurred in Yunnan Province from 2005 to 2015 [46], where the urbanization rate increased from 27.17% in 2005 to 43.33% in 2015, and the IS also increased from 384.1236 km$^2$ in 2005 to 836.6355 km$^2$ in 2015. Based on the demand for sustainable development, in Yunnan Province, the macro-regulation mechanism of land use was strengthened, the structure of land use was actively adjusted, and the efficiency of construction land use was enhanced. In central Yunnan, especially in the Dianchi Basin area, the relationship between population, economic development, and the ecological environment was actively adjusted to build a firm ecological security barrier in the upper reaches of the Yangtze River, which effectively prevented the excessive expansion of the IS.

**Table 6.** Expansion rate of impervious surface in Dianchi Lake Basin.

| Time Range | Rate of Expansion/km$^2$/y | Dilation Degree |
|---|---|---|
| 2000–2005 | 17.38 | Medium-speed expansion |
| 2005–2010 | 36.20 | Fast expansion |
| 2010–2015 | 54.30 | High-speed expansion |
| 2015–2022 | −5.16 | Low-speed contraction |

The rate of change of IS in the main districts and counties in the Dianchi Basin is shown in Table 7. From 2000 to 2005, the IS in the Guandu District expanded at the fastest rate (4.394 km$^2$/y), followed by the Jinning District (3.675 km$^2$/y); Chenggong District expanded at the slowest rate (0.892 km$^2$/y). From 2005 to 2010, the IS in the Xishan District expanded at the fastest rate (16.779 km$^2$/y), followed by the Jinning District (13.972 km$^2$/y); Wuhua District expanded at the slowest rate (1.150 km$^2$/y). From 2010 to 2015, the IS in the Xishan District contracted at a rate of 11.370 km$^2$/y, while the Jinning District had the fastest IS expansion rate (15.961 km$^2$/y), followed by the Chenggong District; Wuhua District still had the slowest expansion rate, but its expansion rate increased slightly relative to that recorded in 2000–2010. From 2000 to 2015, the IS of the Jinning District expanded at a fast rate, and as an important area of the urban "north-south extension" strategy of Kunming, it played a pivotal role in the overall development of Kunming. From 2015 to 2022, the IS of all districts and counties, except for that of the Chenggong District, contracted, and the IS of the Wuhua District contracted at the fastest rate (7.170 km$^2$/y).

**Table 7.** The expansion rate of impervious surface in the Dianchi Lake Basin (km$^2$/y).

| Area | 2000–2005 | 2005–2010 | 2010–2015 | 2015–2022 |
|---|---|---|---|---|
| Wuhua | 2.043 | 1.150 | 3.271 | −7.170 |
| Panlong | 2.853 | 2.439 | 5.042 | −1.757 |
| Guandu | 4.394 | 8.475 | 10.880 | −1.135 |
| Xishan | 2.867 | 16.779 | −11.370 | −0.559 |
| Chenggong | 0.892 | 7.871 | 12.326 | 1.002 |
| Jinning | 3.675 | 13.972 | 15.961 | −1.408 |

4.2.2. The Direction of IS Expansion and Spatial Correlation Analysis

The center of mass of IS shifted to the northwest from 2000 to 2005, to the northeast from 2005 to 2010, to the northeast from 2010 to 2015, and to the northwest from 2015 to 2022 (Figure 6). Overall, with the rapid urban development, from 2000 to 2022, the IS center of mass shifted to the northeast, which indicated that, with the economic and population movement, the IS expanded to the northeast with Dianchi as the center, and the urban core also moved [47]. From the standard deviation ellipse (Figure 6, Table 8), we found that the standard deviation ellipse of the IS of the Dianchi Basin shifted slightly, in the south and north, from 2000 to 2005; the direction changed from 12.95° to 13.78°, the X-axis became shorter, and the degree of dispersion decreased. A shift in the northeast–southwest direction occurred from 2005 to 2010; the direction changed to 13.55°, the X-axis became longer, and the dispersion increased. A shift occurred in the northeast–southwest direction from 2010 to 2015. The direction changed to 12.16°, the X-axis continued to become longer, and the dispersion increased further. From 2015 to 2022, the direction changed to 11.32°, and the X-axis became longer. In general, from 2000 to 2022, the standard deviation ellipse of IS in the Dianchi Basin shifted significantly in the south–north direction, the degree of dispersion continued to increase, and the IS expanded in the north, east, and south.

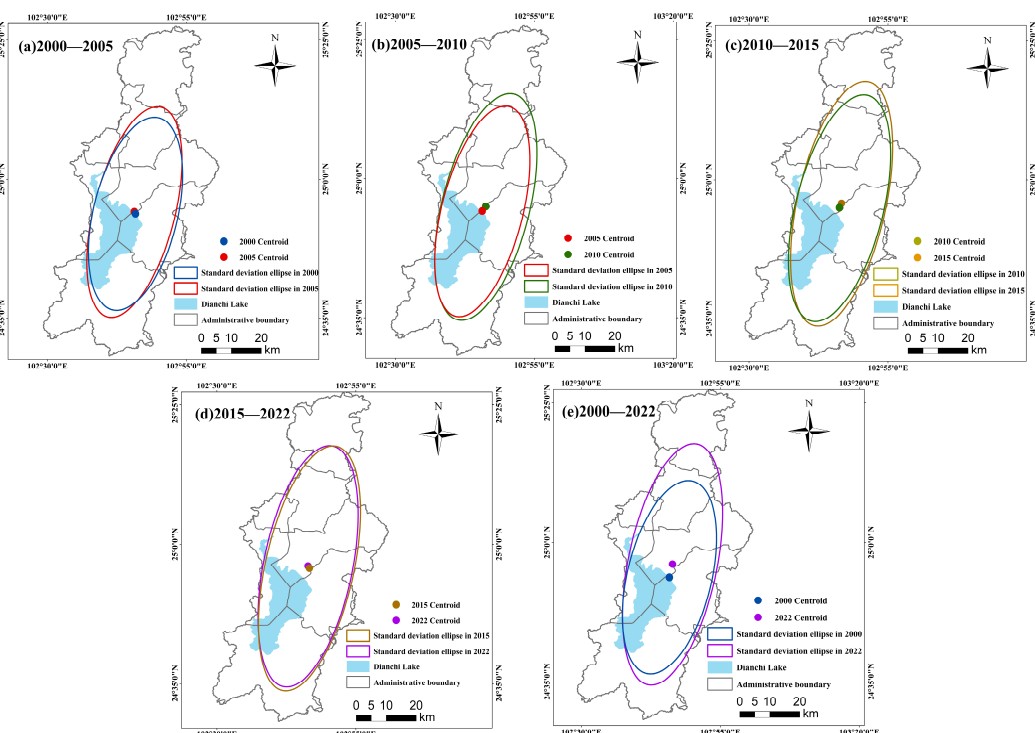

**Figure 6.** IS centroid and standard deviation ellipse transfer direction distribution map.

**Table 8.** Standard deviation ellipse parameters of IS in the Dianchi Lake Basin.

| Year | X-Axis | Y-Axis | X-Axis/Y-Axis | Direction Angle/° |
|------|--------|--------|---------------|-------------------|
| 2000 | 12,827.20 | 32,650.22 | 0.39 | 12.95 |
| 2005 | 12,347.73 | 35,750.50 | 0.35 | 13.78 |
| 2010 | 13,338.70 | 38,401.55 | 0.35 | 13.55 |
| 2015 | 13,462.45 | 41,278.03 | 0.33 | 12.16 |
| 2022 | 13,500.90 | 40,529.79 | 0.33 | 11.32 |

To determine the spatial correlation of IS coverage in the Dianchi watershed from 2000 to 2022, the study area was divided into 1 km × 1 km grids, and the IS coverage on each grid unit was calculated. From the perspective of spatial global autocorrelation, the spatial global Moran's I in 2000, 2005, 2010, 2015, and 2022 were 0.865, 0.847, 0.826, 0.802, and 0.816, respectively, which passed the test at a 1% level of significance, indicating that the IS coverage of the Dianchi Basin, from 2000 to 2022, showed a certain spatial global autocorrelation on the grid scale of 1 km × 1 km. Additionally, we also conducted the hotspot analysis of IS cover in the Dianchi Basin, from 2000 to 2022 (Figure 7), to further determine the local spatial agglomeration characteristics of IS cover in the Dianchi Basin [48]. In 2000 and 2005, there were mainly hotspots and non-significant areas; hotspots were distributed on the east and north sides of Dianchi Lake, and coldspots appeared in 2010. Due to differences in the geographical location and topographic conditions, the coldspots were mainly distributed in Songming County, the north of the Panlong District, the southeast of the Guandu District, and the south and east of the Jinning District, while the hotspots gradually expanded outward from the east and north sides of the Dianchi Lake. This trend matched the expansion trend of the IS.

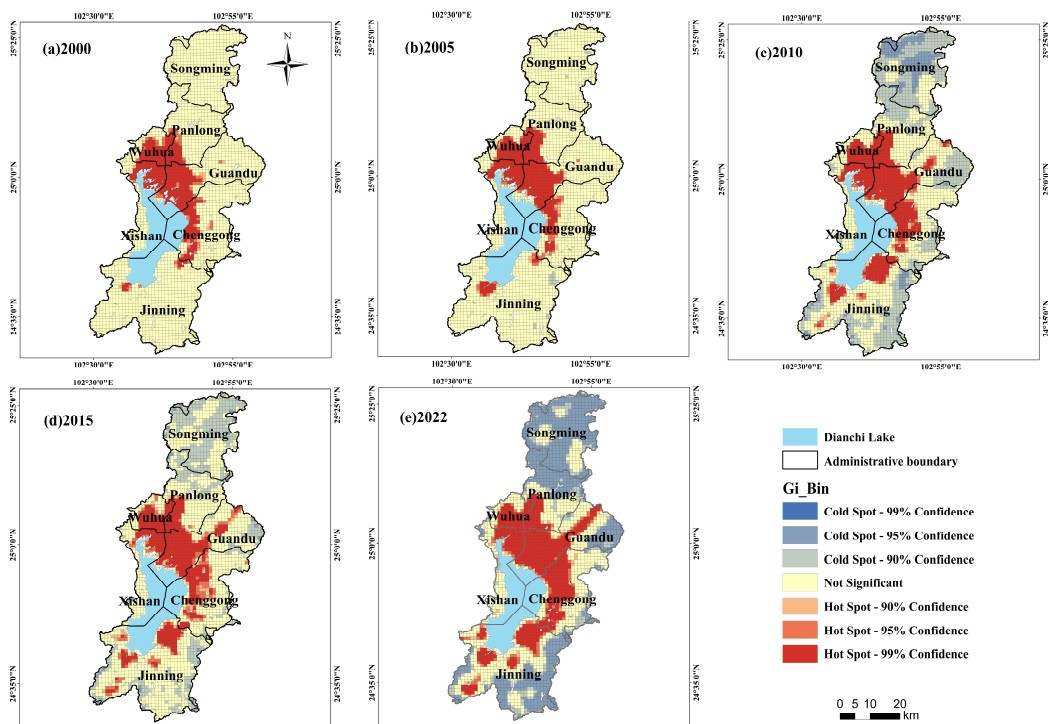

**Figure 7.** Cold and hotspot distribution of IS coverage in the Dianchi Lake Basin from 2000 to 2022.

### 4.2.3. Analysis of Impermeable Surface Drivers Based on PLS-SEM

To determine the accuracy of the model based on PLS-SEM, the variance expansion coefficient (VIF) was used to measure the multicollinearity of multiple variables [43]. The results are shown in Table 9. The VIF of each variable was less than 3, indicating

the absence of multicollinearity among the variables. Additionally, the reliability and validity of the model were tested, and after an appropriate adjustment of each index of the model, the average extracted variation (AVE) was found to be more than 0.5, and the heterogeneity–elemental ratio (HTMT) was less than 0.9. Therefore, the model based on PLS-SEM constructed in this study was reasonable and reliable.

**Table 9.** Variance expansion coefficient of each index in the PLS-SEM model.

| Variable | 2000 | 2005 | 2010 | 2015 | 2022 |
|---|---|---|---|---|---|
| Slop | 1.893 | 1.743 | 1.740 | 1.440 | 1.665 |
| DEM | 2.580 | 2.958 | 2.959 | - | - |
| Mean annual temperature | 1.702 | 2.206 | 2.390 | 2.016 | 2.482 |
| Average annual rainfall | 1.039 | 1.409 | 2.387 | 1.729 | 1.653 |
| GDP | 1.977 | 1.118 | 2.092 | 1.169 | - |
| Population | 1.826 | - | - | 1.215 | 1.351 |
| Distance from the road | 1.146 | 1.118 | - | - | - |
| Distance from scenic spot | - | - | 2.092 | 1.169 | 1.3338 |

The results of the model analysis for 2000–2022 are shown in Table 10. Natural factors negatively affected IS expansion, and this negative effect increased slightly after 2010; the annual average temperature had a suppressive effect on the distribution of IS, and the slope, elevation, and annual average rainfall had a facilitating effect on the distribution of IS. Within the study area, human settlements were mainly concentrated in areas with lower temperatures, and because natural disasters, such as droughts, often occur in Yunnan Province, the IS were mainly concentrated in areas with more rainfall. The elevation and slope were high for mountainous cities, and thus, they facilitated the distribution of IS. Social factors positively affected the distribution of IS, and their effect gradually decreased from 2005 to 2022. Among them, the GDP and population density had a positive effect, while the distance from the road had a negative effect. Since 2010, the index of "distance from the scenic spot" was added to analyze the impact of tourism on the distribution of IS. The "distance from the scenic spot" negatively affected the IS, i.e., the farther away from the scenic spot, the lower the coverage of the IS. This indicated that tourism promoted the expansion of IS.

**Table 10.** The analysis of influencing factors of IS in the Dianchi Lake Basin based on PLS-SEM.

| Impact of Variables on IS Distribution in 2000 | | | | Impact of Variables on IS Distribution in 2005 | | | |
|---|---|---|---|---|---|---|---|
| Natural factors | −0.383 | Average annual rainfall | −0.031 | Natural factors | −0.333 | Average annual rainfall | 0.543 |
| | | Mean annual temperature | −0.740 | | | Mean annual temperature | −0.838 |
| | | DEM | 0.922 | | | DEM | 0.919 |
| | | Slop | 0.864 | | | Slop | 0.834 |
| | | Distance from the road | −0.708 | | | Distance from the road | −0.755 |
| Social factors | 0.398 | Distance from scenic spot | - | Social factors | 0.485 | Distance from scenic spot | - |
| | | GDP | 0.882 | | | GDP | 0.866 |
| | | Population | 0.746 | | | Population | - |

**Table 10.** *Cont.*

| Impact of Variables on IS Distribution in 2010 | | | | Impact of Variables on IS Distribution in 2015 | | | |
|---|---|---|---|---|---|---|---|
| Natural factors | −0.560 | Average annual rainfall | 0.807 | Natural factors | −0.658 | Average annual rainfall | 0.725 |
| | | Mean annual temperature | −0.829 | | | Mean annual temperature | −0.866 |
| | | DEM | 0.827 | | | DEM | - |
| | | Slop | 0.905 | | | Slop | 0.867 |
| | | Distance from the road | - | | | Distance from the road | - |
| Social factors | 0.341 | Distance from scenic spot | −0.907 | Social factors | 0.302 | Distance from scenic spot | −0.749 |
| | | GDP | 0.947 | | | GDP | 0.889 |
| | | Population | - | | | Population | 0.908 |
| Impact of Variables on IS Distribution in 2022 | | | | | | | |
| Natural factors | −0.645 | Average annual rainfall | | | | | 0.578 |
| | | Mean annual temperature | | | | | −0.911 |
| | | DEM | | | | | - |
| | | Slop | | | | | 0.870 |
| | | Distance from the road | | | | | - |
| Social factors | 0.261 | Distance from scenic spot | | | | | −0.889 |
| | | GDP | | | | | - |
| | | Population | | | | | 0.847 |

## 5. Discussion

### 5.1. Effects of IS Expansion on the Ecological Quality in the Dianchi Basin

Greenness, humidity, dryness, and heat are closely associated with the quality of the ecological environment. Most studies used these four indicators to construct the remote sensing ecological index (RSEI) [49–51]. To realize the goal of "carbon neutralization", in this study, we added "carbon storage" to the RSEI. Carbon storage is based on the carbon density dataset and land use type/cover, and it is calculated using the InVEST model. Leveraging insights from prior research [52,53], the carbon density dataset was adapted via the rainfall and temperature model [54,55]. This modification yielded the final carbon density data for the Dianchi Lake Basin. Finally, the improved remote sensing ecological index, C-RSEI, was constructed by conducting a principal component analysis coupled with five indices, including greenness (NDVI), humidity (WET), dryness (NDBSI), heat (LST), and carbon storage (Carbon). A larger value indicated a better eco–environmental quality.

The C-RSEI of the Dianchi watershed in 2000, 2005, 2010, 2015, and 2022 was 0.5054, 0.5071, 0.5118, 0.5085, and 0.5055, respectively, showing an inverted "U" trend, i.e., the eco–environmental quality first increased and then decreased. The temporal and spatial distribution of the eco–environmental quality in the Dianchi Basin, from 2000 to 2022, is shown in Figure 8. The overall eco–environmental quality in the Dianchi Basin showed the zonal distribution characteristics of "good north-south, poor middle". The overall level of urbanization of Songming County, the south of the Jinning District, the east of the Chenggong District, the east and north of the Panlong District, and the eastern part of the Guandu District was low, the intensity of land use development was small, and the quality of the ecological environment was good. In contrast, the quality of the ecological environment was poor in the Wuhua District, the southwest of the Panlong District, the northeast of the Xishan District, the west of the Guandu District, the west of the Chenggong District, and the central part of the Jinning District because of the high level of urbanization, the accumulation of construction land, and the fragmentation of ecological land. From

2000 to 2022, with the continuous expansion of the IS, the eco–environmental quality of the north bank of Dianchi Lake gradually decreased.

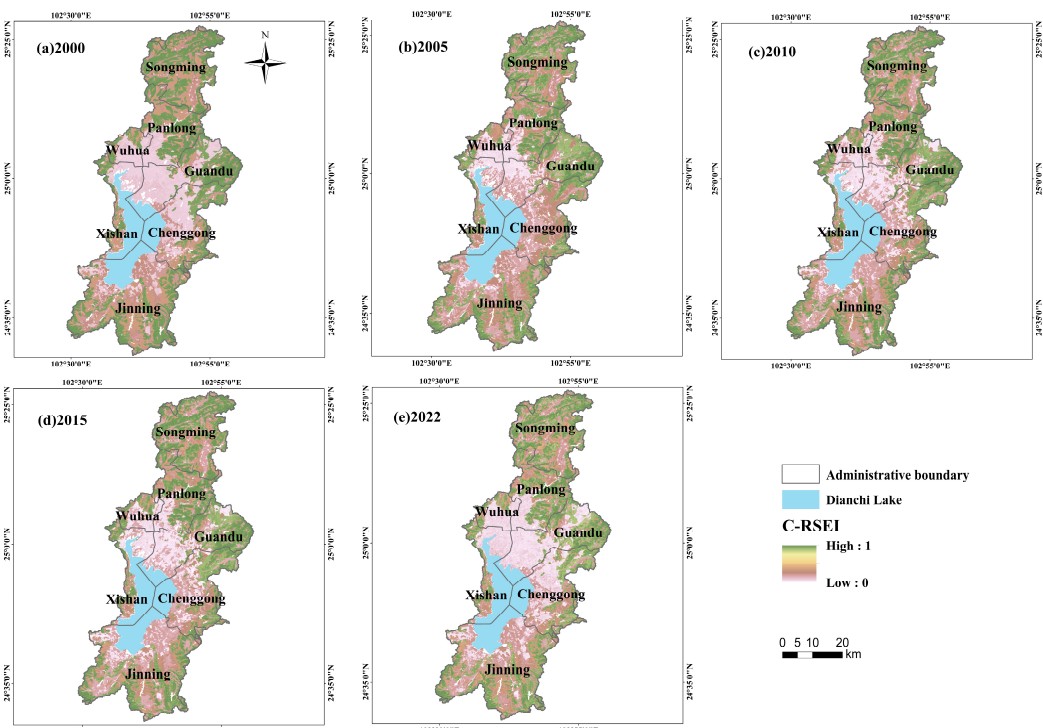

**Figure 8.** The distribution of ecological environment quality in the Dianchi Lake Basin from 2000 to 2022.

To analyze the influence of IS expansion on the quality of the ecological environment, the study area was divided into grids of 1 km × 1 km; then, the IS coverage in each grid was calculated, and the average value of the C-RSEI was extracted. Pearson's correlation coefficients for the relationship between IS coverage and the C-RSEI in the Dianchi Basin, from 2000 to 2022, were −0.408, −0.366, −0.403, −0.419, and −0.532, respectively, indicating a moderate negative relationship between IS coverage and the quality of the ecological environment at the scale of 1 km × 1 km. We then performed the bivariate spatial autocorrelation analysis on the data. The bivariate global spatial autocorrelation was expressed using the bivariate Moran's I index. From 2000 to 2022, the bivariate Moran's I indices of IS coverage and the C-RSEI in the Dianchi Basin were −0.398, −0.354, −0.387, −0.398, and −0.519, respectively, indicating a spatial global negative correlation between IS coverage and the quality of the ecological environment in the Dianchi basin. The bivariate LISA clustering diagram of IS coverage and the C-RSEI, in the Dianchi watershed, is shown in Figure 9. The main clustering types were "low-high" and "high-low", i.e., the quality of the ecological environment of the region with lower IS coverage was better. In contrast, the quality of the ecological environment of the region with higher IS coverage was poor, indicating a spatial local negative correlation between IS coverage and the quality of the ecological environment.

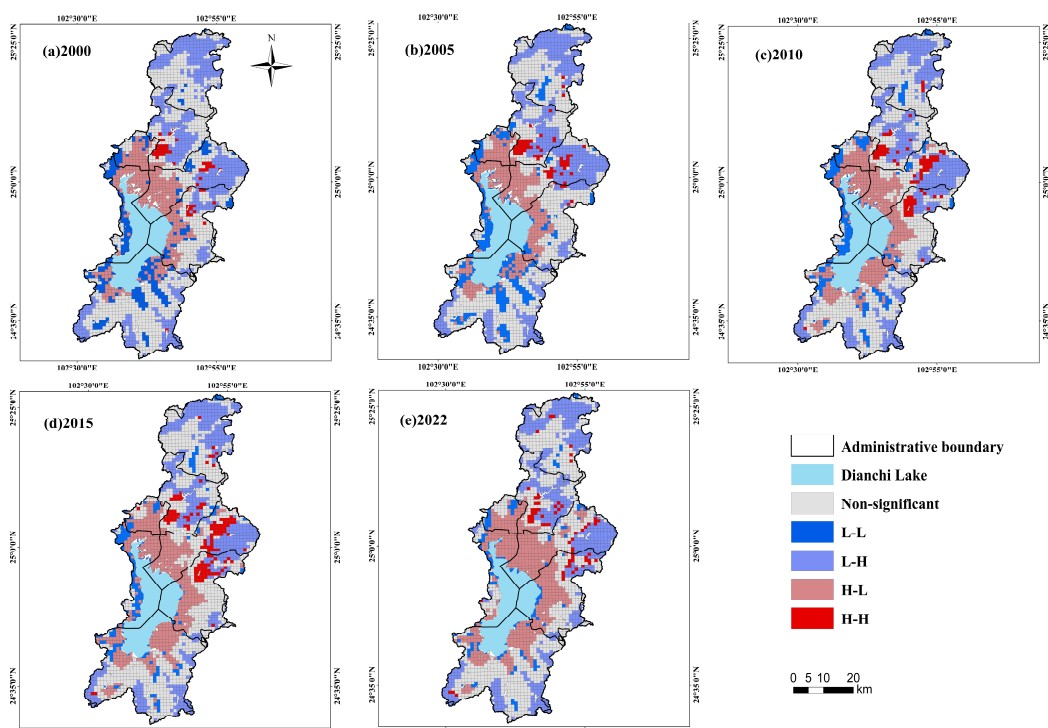

**Figure 9.** Bivariate LISA cluster diagram of impervious surface coverage and ecological environment quality.

*5.2. Comparison of the Results of IS Extraction for Each Coupled Model*

Based on the classification accuracy of each coupling model (Figure 4), in the extraction experiment of ANN, the OA and Kappa coefficients of IMG-SPE-SSC-TEX$^{ANN}$ were the highest (84.0093% and 0.7830, respectively). The UA of IMG-TEX$^{ANN}$ was the highest (92.5602%), and the PA of IMG-SSC$^{ANN}$ was the highest (90.3676%). In general, the classification effect of IMG-SPE-SSC-TEX$^{ANN}$ was the best; the OA and Kappa coefficients, as well as UA of IMG$^{ANN}$, using only images for IS extraction were the lowest, and the PA of IMG-SPE$^{ANN}$ and IMG-TEX$^{ANN}$ were also the lowest. In the extraction experiment based on the SVM coupling model, the OA and Kappa coefficients of IMG-SPE$^{SVM}$ were the highest (91.9841% and 0.8882, respectively). The UA of IMG-SSC$^{SVM}$ was the highest (90.7648%), and the PA of IMG-SSC-SPE$^{SVM}$ was the highest (95.7752%). The extraction effect of IMG-SPE$^{SVM}$ was the best. The OA and Kappa coefficients of IMG$^{SVM}$ were the lowest, the UA of IMG-SPE-SSC-TEX$^{SVM}$ was the lowest, and the PA of IMG$^{SVM}$ and IMG-TEX$^{SVM}$ was the lowest. In the RF-based IS extraction model, the OA and Kappa coefficient of IMG-SSC$^{RF}$ were the highest (90.4579% and 0.8673, respectively). The UA of IMG-TEX$^{RF}$ was the highest (88.7873%). The PA of IMG-SSC-SPE$^{RF}$ was the highest (96.3245%). The OA and Kappa coefficients of IMG-SPE$^{RF}$ were the lowest, the UA of IMG$^{RF}$ was the lowest, and the PA of IMG-TEX$^{RF}$ was the lowest.

We found that only using remote sensing images to extract the impervious water surface limited its extraction accuracy, while using remote sensing image features as auxiliary information improved the classification accuracy to some extent, and the type and fusion number of features affected the extraction accuracy. This was because different kinds of features had the same information, and the fusion of too many features caused redundancy and decreased classification accuracy. For these three machine learning methods, the best overall classification result was provided by the SVM, followed by the RF, and the ANN provided the worst classification result. The coupled models with the highest extraction accuracy among the three machine learning algorithms, including IMG-SPE-SSC-TEX$^{ANN}$, IMG-SSC$^{RF}$, and IMG-SPE$^{SVM}$, respectively, were selected to compare the classification results of the three models.

The differences in the extraction ability of the three IS were mainly concentrated in the extraction of low-reflectivity IS (such as roads, cement floors, etc.), and the details are compared in Figure 10. As shown in Figure 10a, the suburban area in the central part of the Dianchi Lake Basin was where the ANN had low-reflectivity IS leakage, and more IS was erroneously classified as bare land. The ability of SVM and RF to identify low-reflectivity IS was considerably higher than that of ANN. As shown in Figure 10b, at Changshui International Airport in the Guandu District, the ANN also showed IS leakage and identified it as bare land, while RF identified permeable areas (mostly bare land) as IS and showed multi-partitioning identification of the impervious water surface, while the SVM showed a better recognition ability. As shown in Figure 10c, in the eastern suburb of the Chenggong District, it is difficult for ANN to identify IS with low reflectivity, which was similar to the results described in Figure 10a; thus, the ANN was poor at identifying low IS cover areas.

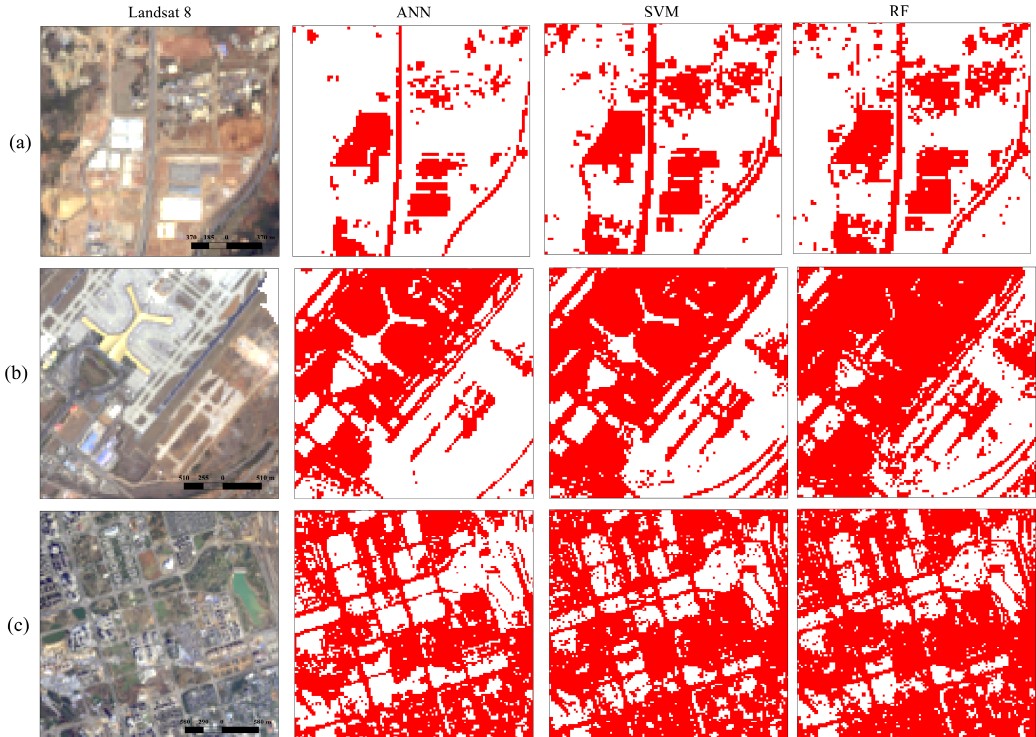

**Figure 10.** Model extraction detail comparison diagram. (**a**) is the suburb of the center of Dianchi Lake Basin. (**b**) is the airport area. (**c**) is the suburb of Chenggong.

### 5.3. Limitations and Prospects

Based on the three machine learning algorithms and the multi-features of coupled remote sensing images, the optimal model was selected for extracting and mapping the impervious water surface of the Dianchi Lake Basin from 2000 to 2022. Starting with the expansion speed, expansion direction, spatial correlation, and driving mechanism, the dynamic characteristics of the IS in the Dianchi Lake Basin, from 2000 to 2022, were analyzed. In this study, we proposed a relatively innovative research framework for extracting and analyzing the IS of regional long-time series. Our methods and findings might be important for the economic and urban development, as well as the sustainable coordination, of the Dianchi Lake Basin.

However, this study had some shortcomings. First, in the IS extraction experiment, the quantity and quality of sample points were directly related to the IS extraction accuracy, so the selection of sample points was very important but extremely tedious. For subsequent experiments on the extraction of IS in a large area, we can use OpenStreetMap and other open-source products to obtain sample points automatically. Second, in this study, only

three machine learning algorithms were used—ANN, SVM, and RF—which have some limitations. We need to use other machine learning algorithms to compare and analyze the results obtained here. Additionally, this study was based on the empirical analysis method, where PLS-SEM was used to identify the driving mechanism of IS expansion in the study area; only the influences of elevation, slope, temperature, rainfall, GDP, population, tourism, road, and other factors were considered, while the reasons for IS changes were complex and diverse. Future studies might consider the impact of natural factors, such as soil types and solar radiation, as well as social factors, such as hospitals and schools.

## 6. Conclusions

Based on the Landsat images from 2000 to 2022, the optimal coupling model was used to extract and analyze the impervious water surface of the Dianchi Lake Basin. The results showed the following: (1) By comparing the confusion matrix and accuracy evaluation results of 24 sets of impervious datasets, the optimal coupling model for extracting IS in the Dianchi Basin was found to be IMG-SPE$^{SVM}$, and the extraction effect of SVM was better than that of the other two machine learning methods. (2) The mapping results of IS, in different years, showed that significant changes occurred in the spatial distribution and shape of IS in the Dianchi Lake Basin between 2000 and 2022, but all changes occurred in the area around Dianchi Lake. (3) The IS expanded at a medium speed from 2000 to 2005, at a fast speed from 2005 to 2010, and at a high speed from 2010 to 2015. The rate of IS expansion showed a sequential acceleration from 2000 to 2015 and contracted slowly from 2015 to 2022. (4) The center of mass of the IS moved to the northeast in 2000, the IS expanded to the northeast with Dianchi Lake as the center, and the urban core also moved with it. The standard deviation ellipse shifted considerably in the south–north direction, and the degree of dispersion continued to increase. The IS expansion showed "north extension, east extension, and south extension". (5) The coverage of IS showed a certain spatial global and local autocorrelation from 2000 to 2022. (6) Natural factors negatively affected the expansion of the IS, and this effect increased slightly after 2010. In contrast, social factors positively affected the distribution of the IS, and its effect gradually weakened from 2005 to 2022. (7) Within the scale of 1 km × 1 km used for the survey, a moderate negative correlation was recorded between IS coverage and the eco–environmental quality in the study area, and a global and local negative correlation was found between them. (8) In this investigation, sample selection largely hinged on visual interpretation, which presents certain limitations. Future studies focusing on impervious extraction might benefit from utilizing open-source products, such as OpenStreetMap, to acquire samples automatically. Further research could also consider deploying a range of machine learning or deep learning algorithms, beyond ANN, SVM, and RF, for comparative evaluation. Moreover, the influence of factors such as soil types, solar radiation, hospitals, schools, and others on the distribution of impervious surfaces can be explored in forthcoming studies.

**Author Contributions:** Conceptualization, X.Y. and Y.L.; methodology, X.Y.; software, X.Y.; validation, B.W., J.Z. and Y.L.; formal analysis, X.Y.; investigation, Y.L.; resources, X.D.; data curation, B.W.; writing—original draft preparation, X.Y.; writing—review and editing, Y.L.; visualization, J.Z.; supervision, Y.L.; project administration, X.D.; funding acquisition, Y.L. All authors have read and agreed to the published version of the manuscript.

**Funding:** This research was funded by Ministry of Education Industry-University Cooperation Education Project, grant number 202102245027 (202102136010), and by Research on identification, monitoring and early warning of major geological hazards in alpine canyon area with 'sky and ground' coordination, grant number 2019FY003017 (K26202000920).

**Data Availability Statement:** The datasets generated or analyzed during the study are available from the corresponding author on reasonable request. These data mainly come from Data Center for Resources and Environmental Sciences of the Chinese Academy of Sciences (https://www.resdc.cn/), Geospatial Data Cloud Official Website (https//www.gscloud.cn/), United States Geological Survey

official website (https://www.usgs.gov/), NASA official remote sensing data network (https://ladsweb.modaps.eosdis.nasa.gov/).

**Acknowledgments:** The authors gratefully acknowledge the support of their families and teachers to conduct this comprehensive study.

**Conflicts of Interest:** The authors declare no conflict of interest.

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
