# Peer review of "Impervious Surface Mapping Based on Remote Sensing and an Optimized Coupled Model: The Dianchi Basin as an Example"

_land, doi:10.3390/land12061210_

Round 1
Reviewer 1 Report
Remote Sensing Dynamic Change of IS Based on an Optimized Coupled Model: The Dianchi Basin as an Example
In this manuscript, the authors mapped the IS of the Dianchi Lake Basin from 2000 to 2022 and determined the dynamic changes. By using Landsat imagery from 2000 to 2022. . Their results showed the following: (1) The optimal model for IS extraction in the Dianchi Lake Basin was IMG-SPESVM based on the support vector machine, remote sensing images, and spectral features. (2) From 2000 to 2022, the spatial distribution and shape of the IS in the Dianchi Lake Basin changed significantly, but they all developed in the area around Dianchi Lake. (3) From 2000 to 2015, the rate of expansion of IS accelerated, while from 2015 to 2022, it showed contraction. (4) From 2000 to 2022, the IS expanded to the northeast. The standard deviation ellipse shifted greatly in the south-north direction. (5) Natural factors negatively affected the expansion of IS, while social factors positively affected the distribution IS.
This manuscript is interesting and it is well presented. I can recommend for publication once the following concerns are satisfactorily addressed:
1. The abstract is not clear and it is not clearly presented. I recommend to the authors review and summarize much better and clearly the main ideas behind this paper.
2. Just the same with the tittle. I also recommend to avoid to use the acronym IS because it could make many readers who are experts on remote sensing but not familiar with IS, could not get the point.
3. In the abstract, but also in the manuscript, I have observed the use of reiterations and expressions that should be improved. Just for example: “In this article, the authors mapped the IS of the Dianchi Lake Basin from 2000 to 2022 and determined the dynamic changes of IS”, where IS is repeated and it could be expressed in a more simple way such: “In this article, our aim is to map the evolution of the impervious surface of the Dianchi Lake Basin by controlling/checking their dynamic changes”
4. Another example: “Along with the different combinations of spectral characteristics, texture characteristics, and seasonal characteristics of land cover…”.-> Try to avoid the excessive use of “characteristics” by using similar words such “features”. After that, “the optimal coupling model for the research area was obtained by coupling with 3 machine learning algorithms, respectively.” -> The authors should use “three” instead of “3”. Also they should use “study area” instead of “research area”.
5. In the introduction, the manuscript argues “The expansion of impervious surface can drive global land cover and land use change and is a result of global economic growth and environmental changes [1]. Impervious surface (IS) is a type of surface coverage where water cannot infiltrate below the surface layer, and it mainly includes artificial landscapes, such as roads, squares, parking lots, and building tops [2].” (…) Some lines after: “As an artificial land cover, the impervious surface strongly affects the quality of the regional ecological environment. The expansion of IS reflects urbanization. With rapid global urbanization, the composition and nature of the underlying land surface have changed, causing many ecological and environmental problems [3-6], such as deterioration of the aquatic environment, enhancement of the heat island effect, aggravation of the risk of urban waterlogging, and the reduction of carbon storage and biodiversity.”-> My recommendation is to improve the argument and extend a little bit because this paragraph is justifying your research. Some arguments could be the unequal urbanization process across the globe or the relationship between global economy/urbanization/environmental costs. In the paper entitled, “Globalization and the shifting centers of gravity of world's human dynamics” the authors discuss exactly this point by evaluating the traces from different global dynamics (urbanization, wealth, population, environmental costs) showing how the impact of Eastern Asian countries is ruling this traces at a global scale. Because your study area is also focused on China, I would suggest you to extend this discussion. Also, I can recommend you some recent studies related to environmental sustainability that comment on this particular case and show you how your discussion could be focused: “In the last few decades, it has become more evident that natural environments are increasingly stressed, potentially harming human communities even in the short to medium term. At a global scale, humanity consumes natural resources 1.8 times faster than the rate at which those resources are generated.” (extracted from Menendez et al. 2023: The nexus between innovation and environmental sustainability)
6. Also in the introduction the authors argue “The machine learning algorithm is an advanced method used for studying IS extraction.” -> I would not deep particularly in machine learning algorithms for studying IS extraction. At least, initially, I would expand this literature review by referring to methods such as the one shown there “The model can avoid the influence of subjective factors in the learning process, and the recognition process is fast and accurate. Sulaiman et al. [14] used the random forest algorithm to extract sample points and artificial neural networks to identify IS. Based on Sentinel 1 and Sentinel 2 data, Shrestha et al. [15] used the random forest algorithm to identify the IS of nine cities in Pakistan. Esch et al. [16] used the support vector machine method combined with Landsat images and road network data to map IS in some parts of Germany”, and after that, show the potential of machine learning algorithms. In the particular case of methods, I suggest to expand the list show more relevant studies focused on land use/land cover’s change detection such for example Hermosilla et al. “Using street based metrics to characterize urban typologies”.
7. About the graphical part,
a. I recommend to include the spatial scale in the sample areas in Figure 3.
b. I recommend to include titles in the X and Y axis and also expressing the units in Figure 4.In Figures 5,6, and 7 the authors must be more consistent in the usage of the spatial scale.
c. Figure 8 shows a very low resolution and it is very difficult to realize the text.
Figure 4.
d. In Figure 11, the authors must include a spatial scale.
Author Response
Dear Reviewer,
Thanks very much for taking your time to review this manuscript. I really appreciate all your comments and suggestions! Please find my itemized responses in below and my revisions/corrections in the re-submitted files.
Point 1:The abstract is not clear and it is not clearly presented. I recommend to the authors review and summarize much better and clearly the main ideas behind this paper.
Response 1: Thank you for your comment. According to your suggestion, we have reviewed and summarized the main ideas of this article more clearly and revised the contents of the abstract section. (lines 13-27)
Point 2:Just the same with the tittle. I also recommend to avoid to use the acronym IS because it could make many readers who are experts on remote sensing but not familiar with IS, could not get the point.
Response 2: We are extremely grateful for your suggestion. We have changed the abbreviation of the title to the full name. And checked the other acronyms in the text to avoid difficult to understand.
Point 3: In the abstract, but also in the manuscript, I have observed the use of reiterations and expressions that should be improved. Just for example: “In this article, the authors mapped the IS of the Dianchi Lake Basin from 2000 to 2022 and determined the dynamic changes of IS”, where IS is repeated and it could be expressed in a more simple way such: “In this article, our aim is to map the evolution of the impervious surface of the Dianchi Lake Basin by controlling/checking their dynamic changes”.
Response 3: Thank you very much for your suggestion. We have revised this sentence to: our objective is to map the evolution of IS in the Dianchi Lake Basin from 2000 to 2022 and analyze its dynamic changes (lines 18-19). And we checked and modified other sentences like this.
Point 4: Another example: “Along with the different combinations of spectral characteristics, texture characteristics, and seasonal characteristics of land cover…”.-> Try to avoid the excessive use of “characteristics” by using similar words such “features”. After that, “the optimal coupling model for the research area was obtained by coupling with 3 machine learning algorithms, respectively.” -> The authors should use “three” instead of “3”. Also they should use “study area” instead of “research area”.
Response 4: Thank you for your comment. We changed "spectral characteristics, texture characteristics, and seasonal characteristics of land cover" to "spectral features, texture features, and seasonal features of land cover" in the full text, and used "three machine learning algorithms" instead of "3 machine learning algorithms" (line 16), and use "study area" instead of "research area".
Point 5: In the introduction, the manuscript argues “The expansion of impervious surface can drive global land cover and land use change and is a result of global economic growth and environmental changes [1]. Impervious surface (IS) is a type of surface coverage where water cannot infiltrate below the surface layer, and it mainly includes artificial landscapes, such as roads, squares, parking lots, and building tops [2].” (…) Some lines after: “As an artificial land cover, the impervious surface strongly affects the quality of the regional ecological environment. The expansion of IS reflects urbanization. With rapid global urbanization, the composition and nature of the underlying land surface have changed, causing many ecological and environmental problems [3-6], such as deterioration of the aquatic environment, enhancement of the heat island effect, aggravation of the risk of urban waterlogging, and the reduction of carbon storage and biodiversity.”-> My recommendation is to improve the argument and extend a little bit because this paragraph is justifying your research. Some arguments could be the unequal urbanization process across the globe or the relationship between global economy/urbanization/environmental costs. In the paper entitled, “Globalization and the shifting centers of gravity of world's human dynamics” the authors discuss exactly this point by evaluating the traces from different global dynamics (urbanization, wealth, population, environmental costs) showing how the impact of Eastern Asian countries is ruling this traces at a global scale. Because your study area is also focused on China, I would suggest you to extend this discussion. Also, I can recommend you some recent studies related to environmental sustainability that comment on this particular case and show you how your discussion could be focused: “In the last few decades, it has become more evident that natural environments are increasingly stressed, potentially harming human communities even in the short to medium term. At a global scale, humanity consumes natural resources 1.8 times faster than the rate at which those resources are generated.” (extracted from Menendez et al. 2023: The nexus between innovation and environmental sustainability).
Response 5: We are extremely grateful for your suggestion. According to your suggestion, we have carefully read and analyzed the two papers you recommended to us, also referred to other related papers, and improved and expanded the argument that " Impervious surface has a strong impact on the quality of regional ecological environment". The argument was extended to the relationship between impervious surface and economic globalization /urbanization, so as to discuss the unequal relationship between economic globalization /urbanization and environmental costs. The improved content corresponds to lines 35 to 48 in the text.
Point 6: Also in the introduction the authors argue “The machine learning algorithm is an advanced method used for studying IS extraction.” -> I would not deep particularly in machine learning algorithms for studying IS extraction. At least, initially, I would expand this literature review by referring to methods such as the one shown there “The model can avoid the influence of subjective factors in the learning process, and the recognition process is fast and accurate. Sulaiman et al. [14] used the random forest algorithm to extract sample points and artificial neural networks to identify IS. Based on Sentinel 1 and Sentinel 2 data, Shrestha et al. [15] used the random forest algorithm to identify the IS of nine cities in Pakistan. Esch et al. [16] used the support vector machine method combined with Landsat images and road network data to map IS in some parts of Germany”, and after that, show the potential of machine learning algorithms. In the particular case of methods, I suggest to expand the list show more relevant studies focused on land use/land cover’s change detection such for example Hermosilla et al. “Using street based metrics to characterize urban typologies”.
Response 6: Thank you very much for your suggestion. In the previous literature review, we only discussed the application of machine learning algorithms in IS extraction, ignoring the application in urban built-up areas and land use change monitoring. Therefore, we have added several studies to prove the significant role of machine learning algorithms in land use and artificial surface monitoring. (lines 62-68 and lines 78-80)
Point 7: About the graphical part,
a. I recommend to include the spatial scale in the sample areas in Figure 3.
b. I recommend to include titles in the X and Y axis and also expressing the units in Figure 4. In Figures 5,6, and 7 the authors must be more consistent in the usage of the spatial scale.
c. Figure 8 shows a very low resolution and it is very difficult to realize the text.
d. In Figure 11, the authors must include a spatial scale.
Response 7: We are extremely grateful for your suggestion.
a. We have added the spatial scale to the sample areas in Figure 3.
b. We added titles to the X and Y axes of Figure 4. There are no definite unit for the Kappa coefficient, so we only added the units of OA/UA/PA. In addition, we unified the spatial scales of Figures 5, 6, 7, 8 and 9.
c. Due to the limitation of the software, it is impossible to export the high-resolution figure. So we displayed the result in the form of a table (Table 10), which is also easy for readers to understand and solves the problem that it is difficult to realize the text.
d. We have added the spatial scale in Figure 11 (Figure 10 now).
We would like to take this opportunity to thank you for all your time involved and this great opportunity for us to improve the manuscript. We hope you will find this revised version satisfactory.
Very sincerely yours, best wishes, best regards.
The Authors

Reviewer 2 Report
In this paper, the authors design a machine learning framework to monitor the impervious surface in Dianchi Lake Basin. On one hand, spectral, texture and seasonal characteristics in the research area are employed to constitute the feature matrix. On the other hand, the program applies neural network, support vector machine and random forest to realize the supervised learning task. It is my understanding that the research article is well organized and the basic idea is reasonable. However, this manuscript should be major revised. My suggestions are listed as follows.
Major Comments:
1. In Section 3.1.4, the authors discuss their experiment results on the machine learning model. However, the control parameters within the machine learning techniques are not explained at all. For example, what is the computer configuration and programming language? How many layers are in the neural network? What is the loss function in neural work? Is the kernel strategy used in the support vector machine? What is the value of relaxation or margin? How many decision trees are in the random forest? What is the feature splitting criterion in the decision tree? It is necessary to explain the implementation details using a new paragraph. The preceding content is important to ensure the reproducibility of the manuscript.
2. In Section 3.1.1, it is necessary for authors to read Wikipedia or other textbooks with the aim of providing an accurate description of three machine learning techniques. For example, the term activation function is more general than the transfer function in Line 161. Moreover, support vector machine is not a binary classification model. It can be used to address the multi-class and regression tasks.
3. In the Conclusion section, it is helpful to discuss the limitation of the current work and present several future directions.
Specific Comments:
Line 1. The abbreviation of the term impervious surface is casual. I recommend the authors did not use the acronym at the title of the manuscript. Moreover, it is favorable to provide the full name and the corresponding acronym when it is first introduced. Therefore, the abbreviation should be mentioned in the first sentence. In addition, the abbreviation is also mentioned in Line 63. It is my suggestion to check the abbreviation issue thoroughly.
Line 37. It is recommended to partition the citation of references 3-6 into the subsequent environmental questions. For instance, “XXX, such as deterioration of environment [3], enhancement of heat island effect [4-5], and aggravation of the risk [6].” The sentence in Line 47 provides a good example.
Line 45. The word poor is not suitable in the academic article.
Line 71. Please provide the full name of NDVI.
Line 87. It is recommended to use the word optimizing, selecting and identifying to replace the verbs. The similar grammar issue should be carefully checked.
Line 100. The superscript is necessary for the number 2 in km2.
Line 109. The legend in Figure 1 is not correct. The gray line does not indicate the boundary of Yunnan Province.
Line 130. There are many unsuitable words in Table 1. Please provide the full name of DEM and GAIA. The first letter in “road data” should be capitalized. Moreover, what are “high efinition satellite images”?
Line 131. It is not proper to use the word “å’Œ” in Table 2.
Line 141. The word characters in Figure 2 is not correct. Based on my interpretation, the word characteristics should be used.
Line 168. What is b(2)? Is it correct?
Line 164. Based on my experiences, it is not suitable to state that SVM is more reasonable and effective than other learning methods in solving small-sample and non-linear problems. Numerous factors could affect the performance of the machine learning methods. SVM is not more powerful than other methods in any problem.
Line 284. The letter M should be italic because it represents a variable. It is required to check the similar issue throughout the manuscript.
Line 361. In Table 6 and the following paragraphs, the unit of the expansion rate is km2/a. What does the letter a mean? Based on Equation 9, T is a time interval or duration. Thus, the unit of T should be year or day.
The English expression should be carefully checked.
Author Response
Dear Reviewer,
Thanks very much for taking your time to review this manuscript. I really appreciate all your comments and suggestions! Please find my itemized responses in below and my revisions/corrections in the re-submitted files.
Point 1:In Section 3.1.4, the authors discuss their experiment results on the machine learning model. However, the control parameters within the machine learning techniques are not explained at all. For example, what is the computer configuration and programming language? How many layers are in the neural network? What is the loss function in neural work? Is the kernel strategy used in the support vector machine? What is the value of relaxation or margin? How many decision trees are in the random forest? What is the feature splitting criterion in the decision tree? It is necessary to explain the implementation details using a new paragraph. The preceding content is important to ensure the reproducibility of the manuscript.
Response 1: Thank you for your comment. The three machine learning algorithms in this paper were implemented by ENVI platform. According to your suggestion, we have described the specific parameters of the three models in detail in Section 3.1.4. (lines 323-335)
Point 2:In Section 3.1.1, it is necessary for authors to read Wikipedia or other textbooks with the aim of providing an accurate description of three machine learning techniques. For example, the term activation function is more general than the transfer function in Line 161. Moreover, support vector machine is not a binary classification model. It can be used to address the multi-class and regression tasks.
Response 2: We are extremely grateful to you for pointing out this problem. We have changed the transfer function of the previous 161 lines to the activation function (line 204). According to your suggestion, we carefully read and learned the basic concepts and principles of the three machine learning algorithms through Wikipedia and textbooks. Three machine learning algorithms were re-described in this paper (lines 190-202 and lines 206-214 and lines 236-239). In addition, Wikipedia describes support vector machines as follows: Given a set of training examples, each marked as belonging to one of two categories, an SVM training algorithm builds a model that assigns new examples to one category or the other, making it a non-probabilistic binary linear classifier (although methods such as Platt scaling exist to use SVM in a probabilistic classification setting). So we modified the description of "support vector machine is a binary classification model" (lines 207-212).
Point 3: In the Conclusion section, it is helpful to discuss the limitation of the current work and present several future directions.
Response 3: We are extremely grateful for your suggestion. We have added the limitations of the current work and the future research direction in the conclusion. This corresponds to lines 663 to 670 in the text.
Point 4: Line 1. The abbreviation of the term impervious surface is casual. I recommend the authors did not use the acronym at the title of the manuscript. Moreover, it is favorable to provide the full name and the corresponding acronym when it is first introduced. Therefore, the abbreviation should be mentioned in the first sentence. In addition, the abbreviation is also mentioned in Line 63. It is my suggestion to check the abbreviation issue thoroughly.
Response 4: We are extremely grateful for your suggestion. We have changed the abbreviation of the title to the full name. And checked the other acronyms in the text to avoid difficult to understand.
Point 5: Line 37. It is recommended to partition the citation of references 3-6 into the subsequent environmental questions. For instance, “XXX, such as deterioration of environment [3], enhancement of heat island effect [4-5], and aggravation of the risk [6].” The sentence in Line 47 provides a good example.
Response 5: We are extremely grateful to you for pointing out this problem. We have divided references 3-6 (5-8 now) into subsequent environmental problems. (lines 44-46)
Point 6: Line 45. The word poor is not suitable in the academic article.
Response 6: We are extremely grateful to you for pointing out this problem. We have modified the word. (line 51)
Point 7: Line 71. Please provide the full name of NDVI.
Response 7: We are extremely grateful for your comment. We have provided a full name for NDVI. (line 93)
Point 8: Line 87. It is recommended to use the word optimizing, selecting and identifying to replace the verbs. The similar grammar issue should be carefully checked.
Response 8: We are extremely grateful for your suggestion. We have modified the verbs. (lines 116-119)
Point 9: Line 100. The superscript is necessary for the number 2 in km2.
Response 9: We are extremely grateful to you for pointing out this problem. We have modified it. (line 136)
Point 10: Line 109. The legend in Figure 1 is not correct. The gray line does not indicate the boundary of Yunnan Province.
Response 10: We are extremely grateful to you for pointing out this problem. We have modified the legend in Figure 1.
Point 11: Line 130. There are many unsuitable words in Table 1. Please provide the full name of DEM and GAIA. The first letter in “road data” should be capitalized. Moreover, what are “high efinition satellite images”?
Response 11: Thank you very much for your suggestion. We have provided the full names of DEM and GAIA in Table 1, and capitalized the first letter in "road data". In addition, "high efinition satellite images" has been modified to "high definition satellite images" because of a spelling mistake. We are very sorry about that. (Table 1)
Point 12: Line 131. It is not proper to use the word “å’Œ” in Table 2.
Response 12: We are extremely grateful to you for pointing out this problem. We have modified it. (Table 2)
Point 13: Line 141. The word characters in Figure 2 is not correct. Based on my interpretation, the word characteristics should be used.
Response 13: We are extremely grateful to you for pointing out this problem. We have changed the word to "features". (Figure 2)
Point 14: Line 168. What is b(2)? Is it correct?
Response 14: We are extremely grateful to the reviewer for pointing out this problem. (2) is the formula number, which was not deleted because of our carelessness, but now we have deleted it. (line 218)
Point 15: Line 164. Based on my experiences, it is not suitable to state that SVM is more reasonable and effective than other learning methods in solving small-sample and non-linear problems. Numerous factors could affect the performance of the machine learning methods. SVM is not more powerful than other methods in any problem.
Response 15: We are extremely grateful to the reviewer for pointing out this problem. We have deleted this section. (lines 206-214)
Point 16: Line 284. The letter M should be italic because it represents a variable. It is required to check the similar issue throughout the manuscript.
Response 16: We are extremely grateful to the reviewer for pointing out this problem. We have corrected it and modified similar problems. (line 363)
Point 17: Line 361. In Table 6 and the following paragraphs, the unit of the expansion rate is km2/a. What does the letter a mean? Based on Equation 9, T is a time interval or duration. Thus, the unit of T should be year or day.
Response 17: Thank you for your comment. "a" stands for year. In order to express the meaning of the unit more clearly, we changed "a" to "y" (lines 441-456). We also emphasized in the Methodology section that "T" represents the year (line 348).
We would like to take this opportunity to thank you for all your time involved and this great opportunity for us to improve the manuscript. We hope you will find this revised version satisfactory.
Very sincerely yours, best wishes, best regards.
The Authors

Reviewer 3 Report
In this paper, impervious surface of a city in China is extracted from remote sensing images by machine learning classifiers. I think the paper is interesting and original, but its structure is not suitable for publications. Please consider following comments to improve its quality:
- Title: please revise it. For example: impervious surface mapping using remote sensing imagery and an optimized model (Case study:)
- Abstract: there are some errors and mistakes in the language. So, please revise it carefully.
- Literature review: the related papers are presented but their achievements or results are not clear.
- Please carefully mention the originality of the manuscript in the end of introduction.
- Data: some data sets are used here, but it is not clear why? Please mention reason behind the use of each data.
- Figure 2: I cannot understand the procedure of the study. Landsat image is the start of the workflow, but I cannot see other data sets. Moreover, some typo errors are observed in the workflow.
- It is very difficult for me to follow concepts in the methodology section. Please present concept step by step according to implementation.
- What is the optimized model? Did you integrate results of classifiers using this mode?
Author Response
Dear Reviewer,
Thanks very much for taking your time to review this manuscript. I really appreciate all your comments and suggestions! Please find my itemized responses in below and my revisions/corrections in the re-submitted files.
Point 1:Title: please revise it. For example: impervious surface mapping using remote sensing imagery and an optimized model (Case study:)
Response 1: We are extremely grateful for your suggestion. According to your suggestion, we modified the title to read: Impervious Surface Mapping Based on Remote Sensing and an Optimized Coupled Model: The Dianchi Basin as an Example.
Point 2:Abstract: there are some errors and mistakes in the language. So, please revise it carefully.
Response 2: Thank you for your comment. We have modified the abstract and asked the professional language editor to modify the language. (lines 13-27)
Point 3:Literature review: the related papers are presented but their achievements or results are not clear.
Response 3: We are deeply grateful to you for pointing out this problem. According to the suggestion of you and other reviewers, we have revised the content of the literature review section. The main conclusions were supplemented to the literature which was lack of conclusion. And some critical discussions and conclusions have also been made on the basis of the display of previous research results. (lines 62-108)
Point 4:Please carefully mention the originality of the manuscript in the end of introduction.
Response 4: Thank you for your suggestion. We have added the innovation of this article at the end of the introduction. This content corresponds to lines 120 to 127 in the text.
Point 5:Data: some data sets are used here, but it is not clear why? Please mention reason behind the use of each data.
Response 5: We are extremely grateful to you for pointing out this problem. We have added the usage of each dataset in Table 1.
Point 6:Figure 2: I cannot understand the procedure of the study. Landsat image is the start of the workflow, but I cannot see other data sets. Moreover, some typo errors are observed in the workflow.
Response 6: Thank you for your comment. We optimized the flow chart in Figure 2 to add the main data to it. (Figure 2)
Point 7:It is very difficult for me to follow concepts in the methodology section. Please present concept step by step according to implementation.
Response 7: We are extremely grateful to the reviewer for pointing out this problem. According to your feedback, we have supplemented the concepts and descriptions of the three machine learning algorithms (lines 190-202 and lines 206-214 and lines 236-239). The formulas of accuracy evaluation methods were added for readers to understand (lines 291-300 and lines 307-310). In addition, the concept of partial least square structural equation model was also added (lines 381-382). The concepts of other methods have existed before, so there are no concepts of adding these methods.
Point 8:What is the optimized model? Did you integrate results of classifiers using this mode?
Response 8: Thank you for your comment. The optimization model proposed in this paper means that different feature combinations are coupled with different machine learning algorithms, and the model with the highest accuracy is selected as the optimal coupling model by evaluating the impervious surface extraction accuracy of multiple coupling models. However, the results of different classifiers are not integrated.
We would like to take this opportunity to thank you for all your time involved and this great opportunity for us to improve the manuscript. We hope you will find this revised version satisfactory.
Very sincerely yours, best wishes, best regards.
The Authors

Round 2
Reviewer 2 Report
Compared with the previous version, the manuscript is significantly improved. The authors diligently modified the method explanation. All my questions have been solved. Therefore, I recommend that this manuscript could be published in its current form.
The English expression should be carefully checked.
Author Response
Dear Reviewer,
Thanks very much for your kind work and consideration on publication of our paper. We have carefully checked the English expression and figures in the paper. On behalf of my co-authors, we would like to express our great appreciation to you.
Very sincerely yours, best wishes, best regards.
Xue Yang
Reviewer 3 Report
-
Author Response
Dear Reviewer,
Thanks very much for your kind work and consideration on publication of our paper. On behalf of my co-authors, we would like to express our great appreciation to you.
Very sincerely yours, best wishes, best regards.
Xue Yang